# *Melanoleuca galbuserae*, *M. fontenlae* and *M. acystidiata*—Three New Species in Subgenus *Urticocystis* (*Pluteaceae*, Basidiomycota) with Comments on *M. castaneofusca* and Related Species

**DOI:** 10.3390/jof7030191

**Published:** 2021-03-08

**Authors:** Vladimír Antonín, Hana Ševčíková, Roberto Para, Ondrej Ďuriška, Tomáš Kudláček, Michal Tomšovský

**Affiliations:** 1Department of Botany, Moravian Museum, Zelný trh 6, CZ-659 37 Brno, Czech Republic; hsevcikova@mzm.cz; 2Via Martiri di via Fani 22, I-61024 Mombaroccio, Italy; r.para@alice.it; 3Department of Pharmacognosy and Botany, Faculty of Pharmacy, Comenius University in Bratislava, Odbojárov 10, SK-832 32 Bratislava, Slovakia; duriska@fpharm.uniba.sk; 4Department of Forest Protection and Wildlife Management, Faculty of Forestry and Wood Technology, Mendel University in Brno, Zemědělská 3, CZ-613 00 Brno, Czech Republic; tomas.kudlacek@mendelu.cz (T.K.); michal.tomsovsky@mendelu.cz (M.T.)

**Keywords:** agaricomycetes, DNA sequencing, systematics, type studies

## Abstract

*Melanoleuca* is one of the taxonomically most complicated genera of Agaricomycetes with several taxonomically lineages. The subgenus *Urticocystis* of the genus *Melanoleuca* contains species with either urticoid or absent cheilocystidia. In this paper, three new European species, *Melanoleuca galbuserae*, *Melanoleuca fontenlae*, and *Melanoleuca acystidiata* are described as new to science. *Melanoleuca galbuserae*, related to *Melanoleuca stepposa* and *Melanoleuca tristis*, was discovered in alpine grasslands in North Italy. The type specimens and recent collections of *Melanoleuca angelesiana*, *Melanoleuca castaneofusca*, *Melanoleuca luteolosperma*, *Melanoleuca pseudopaedida*, and *Melanoleuca robertiana* were sequenced and morphologically examined. Moreover, the related *Melanoleuca microcephala* and *Melanoleuca paedida* were included in morphological examination and DNA sequence analyses. All the species were delimited by macro- and micromorphological characters and the multigene phylogenetic analyses of a combined (ITS, rpb2, and tef1) dataset on the basis of the species tree estimation. In accordance with new molecular and morphological data, we suggest taxonomic reappraisal of *M. pseudopaedida* and *M. robertiana*, and *M. fontenlae* and *M. acystidiata* are proposed as new species. The differences between the type material of *M. angelesiana* from the USA and European *M. angelesiana* specimens are discussed.

## 1. Introduction

*Melanoleuca* Pat. is an agaric genus with about 420 validly published names (http://www.indexfungorum.org, accessed on 26 April 2020) and around 60 species accepted worldwide [1]. This genus is taxonomically extremely complicated because of the overlapping macro- and micromorphological characters among the most of species (e.g., [2,3]). Recently, two subgenera were phylogenetically confirmed, subgen. *Melanoleuca* with well-developed macrocystidia, and subgen. *Urticocystis* Boekhout with either urticoid or absent cheilocystidia [4]. Section *Cognatae* with macrocystidia represents the only one exception in subgen. *Urticocystis* [4].

This study represents a continuation of papers dealing with European taxa of *Melanoleuca* subgen. *Urticocystis* [2,3,5]. During mycological field excursions within the “43° Comitato Scientifico Provinciale” held in Solda (Sulden), Bolzano—South Tyrol, Italy, in July 2018, and organized by the Bolzano group of the Associazione Micologica Bresadola, basidiomata of remarkable *Melanoleuca* were found in the alpine grassland with a dwarf willow (*Salix herbacea*). According to its macro- and micromorphological characters and the molecular data, it belongs to subgen. *Urticocystis* and represents a new species described here.

*Melanoleuca castaneofusca* Contu is a lesser-known species from Sardinia, Italy. The species has been known only from the type locality, but according to our records, it is distributed across several European countries. *Melanoleuca castaneofusca* is related to *Melanoleuca luteolosperma* (Britzelm.) Singer, *Melanoleuca microcephala* (P. Karst.) Singer, and *Melanoleuca paedida* (Fr.) Kühner & Maire. The subject of this paper is detailed taxonomical revision of these species including available type collections and recent specimens. The study is based on a macro- and micromorphological examination and a multigene phylogeny of both recent and type specimens. The information about ecology and distribution of all studied species are summarized here.

## 2. Materials and Methods

### 2.1. Morphological Dataset

Macroscopic descriptions, on the basis of fresh basidiomata, were made by the authors or collectors. Color abbreviations follow Kornerup and Wanscher [6], Munsell [7], and Küppers [8]; herbarium abbreviations follow Thiers [9]. Authors of fungal names are cited according to the Authors of Fungal Names page (http://www.indexfungorum.org/AuthorsOfFungalNames.htm, accessed on 5 February 2021). The so-called finger test is used for the character of the pileus surface [10]. A positive test means that the finger leaves a clear imprint on the pileus surface. The caulohymenium, formed in particular on the stipe apex surface in some *Melanoleuca* species, is a layer composed of caulobasidioles, caulocystidia, and sporulating caulobasidia. It is comparable with the hymenium of the hymenophore in many respects [11]. Microscopic features were described from dried material mounted in KOH, Melzer’s reagent, and Congo Red, using an Olympus BX-50 light microscope (Tokyo, Japan), Leica DM 1000 (Wetzlar, Germany), and Olympus BX41 with a magnification of 1000×. The SEM microphotograph of basidiospores was taken using a Tescan Mira 3 LMU electron microscope (Brno, Czech Republic). For basidiospores, the factors E (quotient of length and width in any one spore) and Q (mean of E-values) were used. Characters of cheilocystidia are defined according to Vizzini et al. [4] and Boekhout [12]. The notation [a/b/c] at the beginning of micromorphological data means (a) structures were measured from (b) basidiomata taken from (c) collections.

### 2.2. Molecular Dataset

DNA from dried fungal material was isolated, and DNA of 3 genes (ITS region of ribosomal RNA gene = ITS; translation elongation factor 1-alpha = tef1; RNA polymerase II, the second largest subunit = rpb2) was amplified in accordance with the work of Antonín et al. [2,3]. In the case of older type specimens, the genus-specific primers for the *Melanoleuca*-targeting ITS2 region (MELITS2F/MELITS2R) developed by Antonín et al. [2] were applied for amplification. The 2 datasets were the subject of phylogenetic analyses. The ITS-only dataset contained of all specimens including the type specimens, whereas the combined multilocus ITS–tef1–rpb2 dataset contained selected representatives of each species. The sequences were aligned using MAFFT, version 7 online program, setting up the Q-INS-i option [13].

The datasets were enriched with sequences published by [2,3,4,5,14,15,16]. The aligned ITS dataset was 780 bp long (87 conserved, 74 variable, and 33 singleton positions, as determined in the MEGA X program version 10.1.8 [17]). The combined ITS–tef1–rpb2 dataset was 2748 bp long (1750 conserved, 959 variable, and 98 singleton positions). The DNA sequences of *Melanoleuca friesii* and *Melanoleuca strictipes* published by Ďuriška et al. [18] and Antonín et al. (in prep.) from the subgenus *Melanoleuca* were selected as the outgroup for both ITS only and multilocus datasets.

### 2.3. Phylogenetic Analysis

Phylogeny for both datasets, multilocus (tef1, rpb2, and ITS) and ITS only, was inferred by means of the maximum likelihood and Bayesian methods.

Within the maximum likelihood analysis, we estimated best-fitting partitioning schemes and evolutionary models for each subset with PartitionFinder 2 [19] according to the corrected Akaike Information Criterion (AICc). For both datasets, the analysis was carried out for both linked and unlinked branch lengths and the results were exactly the same in the case of both datasets. The program was set to test the largest set of models possible, including models with base frequencies estimated using maximum likelihood (84 models in total, the option models = allx), and all possible partitioning schemes (the option search = all).

Phylogenetic inference estimated on the basis of the maximum likelihood method was produced in the RAxML-NG software [20]. The sufficient number of bootstrap replicates was determined automatically using the MRE-based bootstopping test [21] with the cutoff value 0.01 (the option—bs-cutoff 0.01). The maximum number of replicates was set to 5000 (the option—autoMRE{5000}). When the convergence was not reached, the analysis was run again for another 5000 bootstraps, both log files were concatenated, and the convergence was tested post hoc using the command bsconverge. We repeatedly added 5000 replicates until the convergence was reached. For the multilocus dataset, the bootstopping test converged after 6200 replicates. For the ITS only dataset, the convergence was not reached even after 30,000 for the cutoff value 0.01, but converged after 1700 trees for the cut-off value 0.03.

We calculated 2 support metrics: (1) Felsenstein’s bootstrap and (2) Transfer Bootstrap Expectation [22]. The latter better revealed support for some branches, and it is displayed in the final trees.

Bayesian inference was computed using the BEAST 2 software [23]. For the ITS only dataset, the analysis was carried out using the Standard template. For the multilocus dataset, the species tree inference was performed via the StarBEAST package [24]. Partitioning schemes selected by PartitionFinder2 (see above) were used as an input for both of these analyses. In both analyses, the site models were chosen automatically through the model averaging implemented in the bModelTest package [25]. The posterior estimates of the parameters were summarized with Tracer [26]. The quality of posterior estimates was evaluated on the basis of estimated sample size (ESS) value and visual analysis of the trace plots. In the case of the ITS only dataset, we used the Markov chain Monte Carlo (MCMC) chain length of 50,000,000, which led to all parameters showing evidence of thorough sampling (ESS >> 1000). For the StarBEAST analysis, the number of MCMC generations resulting in sufficient sampling was much higher—we gradually increased the chain length to up to 400,000,000 when the ESS for all parameters was high enough. All estimates had ESS > 200, and the majority of them >> 1000. Both analyses were set up to sample both the tree and trace every 5000th state. Operators were always adjusted according to the program’s suggestions in the output of the previous run. As a consensus tree method, we opted for the maximum clade credibility tree and produced it using the TreeAnnotator program (a part of BEAST 2). Burn-in was set to 25% and the height of each node to the mean height across the entire sample of trees for that clade.

## 3. Results

### 3.1. Molecular Dataset and Phylogeny

The phylograms resulted from both maximum likelihood and Bayesian methods had almost identical topologies. Within the Bayesian framework, convergence for both datasets was assessed on the basis of the visual assessment of the trace plot and the estimated sample size (ESS) of the posterior estimates of the parameters; in all cases, the value was higher than 200.

The specimens used in molecular analyses are summarized in Table 1. The phylogenetic analyses of both datasets (Figure 1 and Figure 2) resulted in independent and well supported positions of several *Melanoleuca* spp. insufficiently included in previous phylogenetic studies and one species new to science. We succeeded in ITS sequencing of type specimens of *Melanoleuca angelesiana* A.H. Sm., *Melanoleuca castaneofusca*, *Melanoleuca luteolosperma, Melanoleuca pseudopaedida* Bon, and *Melanoleuca robertiana*. The holotype of *M. pseudopaedida* is conspecific with *M. luteolosperma*, and therefore *M. pseudopaedida* in current concept [3] is described here as a new species, *Melanoleuca fontenlae*. Surprisingly, sequence of *M. robertiana* holotype fell among the subgenus *Melanoleuca* (species with macrocystidia), and therefore a taxonomic solution of this problem was proposed.

### 3.2. Taxonomy

***Melanoleuca galbuserae*** Antonín, Ševčíková, Para & Tomšovský, sp. nov. (Figure 3 and Figure 4). Mycobank MB 838741.

*Diagnosis. Melanoleuca stepposa* differs by a dark brown or gray-brown context in the stipe base; a well-developed caulohymenium; and different sequences of ITS, tef1, and rpb2 genes.

*Holotype.* Italy, Alto Adige (South Tyrol), Solda, Rifugio Milano (Schaubachhütte), 46°29′26″ N, 10°35′58″ E, alt. ≈2630 m a.s.l. (above sea level), 29 July 2018 leg. A. Galbusera and H. Ševčíková (BRNM 825709, GenBank/EMBL: MW491332).

*Etymology.* Named according to the co-collector, Italian mycologist A. Galbusera.

Pileus 15–40 mm broad, ± applanate, usually with low obtuse umbo at center, depressed around it when mature, inflexed at margin, smooth, white or concolorous pruinose or almost glabrous (finger test positive or negative; [10]), whitish to light beige, ochre or pale brown, sometimes with light gray tinges (10YR8/1–2, 7/1–4), with darker center—dark yellow-brown, grayish brown, or dark brown (7.5YR4/2–4, 3/2; 10YR6/4–6/6, 5/3–6, 4/3–4, 3/4). Lamellae moderately close, L = ≈50–60, l = 2–3, emarginate and attached with tooth, sinuate, white, then dirty whitish (lighter than 10YR8/1), with uneven, concolorous edge. Stipe mostly (slightly) shorter that the pileus width, 15–25 × 2.5–5 mm, cylindrical, not or slightly clavate at base, longitudinally innately fibrillose, finely pruinose-pubescent only at apex, ochre to pale brown or gray-brown (10YR7/4–6, 6/6, 5/2–6); paler, whitish when young, at apex. Context white in the whole basidioma or only slightly grayish in the stipe base, with slight earthy smell.

Basidiospores [90/3/3] (7.2)7.5–9.5(10) × 5.0–6.5(7.0) μm, average = 8.5 × 5.7 μm, E = (1.23)1.30–1.75(1.84), Q = 1.50, (broadly) ellipsoid, less frequently ellipsoid–fusoid or obovate, ornamentation of small to moderately large warts and small, sometimes rare ridges, amyloid. Basidia [26/3/3] (28)31–39(42) × (9.5)10–13 μm, four-spored, clavate, or subfusoid. Basidioles (11)15–40 × (4.0)5.0–13 μm, clavate, subcylindrical, rarely subfusoid. Cheilocystidia [25/3/3] (length × base width × apex width) (26)32–47(55) × (5.0)6.0–8.0(12) × 2.5–4.0(6.0) μm, urticoid, of both brevipes- and exscissa-type, thin-walled, basal part clavate, fusoid, subcylindrical, sometimes irregular, apex subulate to conical, sometimes cylindrical, obtuse to subacute, rarely broadly obtuse, rarely metuloid. Pleurocystidia absent. Marginal cells 14–22 × 5.0–10 μm, cylindrical, clavate, often two-celled, sometimes irregular, thin-walled. Trama hyphae cylindrical to (sub)inflated, thin-walled, 3.0–12(20) μm, inamyloid. Pileipellis a (sub)trichoderm, of cylindrical, interwoven, up to 10 μm wide hyphae; terminal cells suberect to erect, rarely adpressed, cylindrical, clavate, fusoid, subutriform, thin-walled, obtuse, 3.0–12 μm wide. Stipitipellis a cutis, of cylindrical, slightly thick-walled, smooth or incrusted, 3.0–6.0 μm wide hyphae. Caulocystidia in (small) groups, 23–45(55) × 5.0–8.0(11) μm, cylindrical, narrowly clavate, fusoid, thin-walled. Clamp connections absent in all tissues.

*Ecology*. On soil in alpine grasslands with *Salix herbacea*.

*Distribution*. Thus far, *M. galbuserae* has been found only in three localities in North Italy, but its occurrence in neighboring alpine countries (Austria and Switzerland) is expected.

*Additional specimens examined*. Italy: South Tyrol, Solda, Refugio Madriccio (Madritschhütte), 46°29′48″ N, 10°36′48″ E, alt. ≈2820 m a.s.l., 29 July 2018 leg. J. Hrabáková (BRNM 825710).—Trentino, Pozza di Fassa, Passo delle Selle, 6 August 1994 leg. E. Bizio (MCVE 4505, as *M. grammopodia*).

*Remarks*. *Melanoleuca galbuserae* is characterized by rather small basidiomata, a light beige to dirty ochre-brown pileus with a darker center, white, then dirty whitish lamellae, a cylindrical, ochre to gray-brown stipe, mostly whitish context, (7.2)7.5–9.5(10) × 5.0–6.5(7.0) μm basidiospores, urticoid cheilocystidia of the brevipes- and exscissa-type, a pileipellis in the form of a (sub)trichoderm and simple caulocystidia (caulohymenium absent). It belongs to the *M. exscissa* (Fr.) Singer group [3].

For the phylogenetically related species, *Melanoleuca stepposa* Vacek has a pileus with more yellow tinges and a dark brown or gray-brown context in the stipe base, slightly narrower basidiospores (7.5–10 × 4.5–6.0(6.5) μm), and a well-developed caulohymenium [3]. *Melanoleuca tristis* M.M. Moser also differs by larger basidiomata (pileus 30–65 mm broad, stipe 22–55 × 3.5–7 mm), a dark brown pileus, grayish to gray lamellae, an uniformly dark gray-brown or black-brown to dirty dark brown stipe, dark brown context in the stipe base, larger (30–56 × 5.0–11 μm) cheilocystidia, and a well-developed caulohymenium [3]. *Melanoleuca griseobrunnea* Antonín, Ďuriška & Tomšovský has a gray-brown pileus, dirty cream-colored lamellae, slightly smaller basidiospores [7.0–9.0(9.5) × 5.0–6.0(6.5) μm], and a pileipellis in the form of a cutis [3]. *Melanoleuca porphyropoda* X.D. Yu has a centrally orange-cinnamon, otherwise light pinkish cinnamon to pinkish cinnamon pileus; a light purplish vinaceous stipe; and larger basidiospores (8.0–12.0 × 4.5–8.0 μm) [13]. *Melanoleuca zaaminensis* Kalamees differs by a dark yellowish brown, basally blackish brown stipe; a blackish brown context in the stipe base; and slightly smaller basidiospores [7.5–8.5(9.0) × 5.0–6.0 μm, average 8.1 × 5.6 μm] [3,27].

#### The Description of the *Melanoleuca Castaneofusca* Group

***Melanoleuca fontenlae*** Para, Antonín, Ďuriška, Ševčíková & Tomšovský, sp. nov. (Figure 5). MycoBank MB 838743 = *Melanoleuca pseudopaedida* sensu Vizzini et al., 2011.

*Diagnosis*. *Melanoleuca luteolosperma* differs by slightly smaller basidiospores, (6.0)7.0–9.0 × (4.0)4.2–5.5(6.5) μm and by different sequences of ITS, tef1, and rpb2 genes.

*Holotype*. Italy, Emilia-Romagna Prov., Ravenna distr., Lido di Dante, 44°23′25″ N, 12°18′50″ E, 9 November 2000 leg. V. Antonín (BRNM 772194, GenBank/EMBL: MW491326—ITS, MW488158—tef1, MW488173—rpb2).

*Etymology*. Named “*fontenlae*” in honor of the Italian mycologist Roberto Fontenla, a longtime collaborator of R. Para in *Melanoleuca* studies.

Pileus 9–39 mm broad, broadly conical, or almost applanate, with indistinct or distinct, broad, conical, obtuse umbo at center, inflexed at margin, hygrophanous, margin smooth when young, later sometimes shortly striate, not pruinose, whitish gray, gray-brown, dark brown (5B2, 6C4, 7D3, 7E6–7, 7F5–6, 8E4–5), pallescent up to ochraceous brown to brown (6C–D4–6, 7D5–6), margin paler than center in young specimens. Lamellae moderately close, L = c. 30–65, l = 2–3, emarginate and attached with tooth, ± horizontal, not intervenose, whitish or cream-colored, then pale ochraceous (4A3) or grayish, with beige reflex, with concolorous, ± smooth, finely pubescent edge. Stipe longer than pileus width, 17–60 × 1.5–7 mm, cylindrical, slightly broadened at apex, clavate (up to 5 mm broad) at base, finely floccose-pubescent, especially at apex, entirely longitudinally fibrillose, lustrous, whitish to slightly brownish at apex, ochraceous yellowish to brownish (6C–D3–4, 7D–C4, 7D5) at center, dark gray-brown (7E–F4, 7E–F6, 8E–F3), with white or whitish basal mycelium. Context whitish, slightly grayish brownish under pileipellis, hollow, fibrillose, brown to dark brown in stipe base, with indistinct or slightly fungoid (earthy) smell and mild taste.

Basidiospores [240/10/10] (6.0)6.5–10 × (4.0)4.8–6.0(6.5) μm, average = 8.0 × 5.6 μm, E = (1.1)1.21–1.6(1.73), Q = 1.42, (broadly) ellipsoid, verruculose, warts mostly irregularly shaped and sized, sometimes up to 0.75 μm high, sometimes with rare ridges, amyloid. Basidia [37/6/6] 24–50 × 7.0–13 μm, four-, rarely two-spored, clavate. Basidioles 15–40 × 5.0–12 μm, clavate, subcylindrical. Cheilocystidia [82/10/9] (length × base width × apex width) 19–65 × 3.0–10 × 2.5–3.5 μm, urticoid of the brevipes- and exscissa-type, sometimes very rare or absent, basal part clavate, subfusoid, often irregular, apical part subulate, obtuse, rarely with two septa, thin-walled, with crystals or not. Marginal cells 15–40 × 4.0–10 μm, clavate, ± cylindrical, subfusoid, irregular or branched, thin-walled. Pleurocystidia absent. Trama hyphae ± cylindrical, fusoid to inflated, thin-walled, inamyloid, up to 15 μm wide. Pileipellis an ixocutis transient to ixo(sub)trichoderm at center; terminal cells adpressed to erect, cylindrical, (sub)fusoid, subclavate, obtuse, thin-walled, inamyloid, up to 50 × 10 μm wide, grayish brown in KOH. Stipitipellis a cutis of cylindrical, parallel, ± slightly thick-walled, inamyloid, up to 6.0 μm wide hyphae. Caulohymenium of (1) single or in groups, 21–50 × 6.0–10 μm, clavate, fusoid to cylindrical, sometimes narrowly cylindrical, 30–40 × 2.5–3.0 μm, with apical incrustation, with or without one septum, and (2) urticoid cystidia, 25–70 × 3.0–10 μm, with apical crystals or not, thin-walled; caulocystidia sometimes scattered to absent. Clamp connections absent.

*Ecology*. Growing on sandy soil and dunes; found in grass, on a path, under *Pinus pinaster*, *Pinus maritima*, *Pinus pinea*, *Quercus ilex*, *Pyracantha*, and *Rubus* and under *Salix* and *Pinus* and under *Juniperus communis*

*Distribution*. Known only to be from France, Italy, and Slovakia until now.

*Additional specimens examined*. France: Bais de Somme, 7 December 1981 leg. Mrs. Bergeron, det. M. Bon (PC, Romagnesi 81.253, as *M. rasilis* var. *pseudoluscina*).—Italy: Emilia-Romagna Prov., Ravenna distr., Pineta di San Vitale, Bardello, 10 November 2000 leg. V. Antonín (BRNM 825711).—Ibid., leg. M. Enderle (BRNM 825712).—Ibid., 9 November 2000 leg. V. Antonín and A. Hausknecht (BRNM 825713).—Ibid., Pineta Ramazzotti and Dunes di Lido di Dante, alt. –14 m a.s.l., 5 November 2007 leg. V. Antonín (BRNM 825714). — Venezia, Caorle, Valle Vecchia‒Brussa, 20 December 2014 leg. E. Campo (BRNM 825715).—Slovakia: Cerová vrchovina, Vlčia dolina, 27 October 2002 leg. K. Skokanová (SAV F-3825, SAV F-3822).—Ibid., 27 October 2004 leg. S. Adamčík (SAV F-3823, SAV F-3824).

*Remarks*. *Melanoleuca fontenlae* is characterized by small basidiomata with a gray-brown pileus pallescent up to ochraceous brown, whitish, or cream-colored, then pale ochraceous or grayish lamellae, a slightly clavate, brownish to dark gray-brown stipe, a brown to dark brown context in the stipe base, urticoid, but sometimes rare or even missing cheilocystidia and mostly well-developed caulohymenium.

*Melanoleuca fontenlae* is identical with *M. pseudopaedida* sensu Vizzini et al. [3]. However, the type specimen of *M. pseudopaedida*, phylogenetically tallies to *M. luteolosperma*. Therefore, we consider *M. pseudopaedida* in the original sense as a synonymum of *M. luteolosperma*.

Among phylogenetically close species, only *Melanoleuca luteolosperma* has similarly small basidiomata. It differs by slightly smaller basidiospores, (6.0)7.0–9.0 × (4.0)4.2–5.5(6.5) μm, average 7.7 × 5.1 μm. *Melanoleuca paedida* differs by a larger, 30–60 mm broad, ochraceous fawn to pale or dark gray-brown pileus; an only slightly darker context in the stipe base; and smaller basidiospores, 6.5–8.5 × 3.9–5.5 μm, average 7.3 × 5.1 μm; moreover, it constantly lacks a caulohymenium. *Melanoleuca castaneofusca* differs by larger basidiomata; a distinctly floccose to floccose-tomentose stipe at apex with an only brownish yellow tinged context in its base; and smaller basidiospores, 6.5–8.0 × 4.0–6.0 μm, average = 7.3 × 5.0 μm. *Melanoleuca microcephala* differs by a larger, (grayish) brown, beige-gray, or grayish pileus and consistently absent cheilocystidia and caulohymenium. *Melanoleuca stridula* (Fr.) Singer has a larger, 18–50 mm broad, rather dark (gray-)brown pileus; a distinctly bulbose stipe; slightly smaller basidiospores, 7.0–8.0(9.0) × 4.75–6.0 µm, average 7.9 × 5.4 µm; and consistently absent cheilocystidia and caulohymenium (our unpublished studies). The European collections of *M. angelesiana* have a larger, 25–80 mm broad pileus; an entirely floccose stipe with a pale context in its base; and constantly absent cheilocystidia and caulohymenium (our unpublished studies, see also notes above). *Melanoleuca acystidiata* has a brown pileus; a whitish, then light brown stipe; a white context in the stipe base; and a lack of cheilocystidia.

***Melanoleuca robertiana* Bon**, Documents Mycologiques 20(79): 58, 1990.

*Type revision* (LIP 72092034, GenBank/EMBL: MW491341): Basidiospores 8.0–9.5(10) × 4.5–5.3 µm, average = 8.85 × 4.85 µm, E = 1.6–2.1, Q = 1.83, ellipsoid, oblong-ellipsoid, ellipsoid-fusoid, ornamentation of ± regular warts variable in size, ridges absent or very rare. Basidia 27–32 × 10–12 µm, four-spored, clavate or subfusoid. Basidioles 20–33 × 6.0–11 µm, clavate, subfusoid. Cheilocystidia 60–69 × 14–18 µm, fusoid or sublageniform, thin-walled with slightly thick-walled apex. Pleurocystidia absent. Trama hyphae cylindrical to ellipsoid, thin-walled, 5.0–15(20) µm wide, inamyloid. Pileipellis a cutis of cylindrical, thin-walled, 4.0–10 µm wide hyphae; terminal cells adpressed to suberect, cylindrical, obtuse, simple, rarely branched. Stipitipellis a cutis of cylindrical, parallel, 3.0–6.0 µm wide hyphae. Caulocystidia 28–41 × 7.0–12 µm, clavate, fusoid, subutriform, thin-walled; one macrocystidium seen, 44 × 13 µm. Clamp connections absent in all studied tissues.

*Remarks*. Bon [28] described *Melanoleuca robertiana* as an acystidiate taxon with basidiospores of (6.5)7–8.5(9) × (5)5.5–6(6.5) µm. The holotype was preserved in the Herbarium M. Bon, deposited in LIP (LIP 72092034). The majority of holotype material was not found in LIP, but a pocket with a small piece of holotype was glued to the original author’s sheet with a description of the holotype, consisting of roughly one-quarter of one small basidioma. Bon mentioned the absence of any cystidia and basidiospores of 6–7.5(8) × 5–6 µm on this description accompanying the holotype material. Macro- and microscopic characters of the *Melanoleuca robertiana* published in the literature [28,29] indicates the possibility this species belonging to the *M. castaneofusca* group. However, our holotype revision revealed the presence of 60–69 × 14–18 µm large cheilocystidia and basidiospores of 8.0–9.5(10) × 4.5–5.3 µm in size. The ITS sequencing of the holotype was successful, and ITS sequence agreed with that of macrocystidioid *M. pallidicutis* Bresinsky holotype (TAAM 178616, MT270846) [30] belonging to the subgenus *Melanoleuca*. The obvious disagreements between these features of the studied piece of the type material and the characters described in the original protologue of *M. robertiana* indicate that the original description and holotype specimen refer to two different taxa. It is almost certain that a mycologist as experienced as M. Bon described in the protologue a different collection than that which represents the type material of *M. robertiana*. In the literature, *M. robertiana* has always been considered an acystidiate species [4,10,31,32] (the last as synonym of *M. melaleuca*). We are convinced that the material was mistakenly confused and a wrong basidioma was deposited in the herbarium envelope. *Melanoleuca pallidicutis* is a taxon based on an unambiguous well-defined description and whose morphological characters in protologue match with that of the holotype and the DNA sequence supports its expected taxonomic position (Antonín et al. in prep.). On the basis of these facts, we considered the name *Melanoleuca robertiana* Bon a *nomen confusum* because there is a substantial conflict in crucial characters (the presence/absence and character of cystidia and spores size) between descriptions published in the protologue, other literature [10,28,29], and the type specimen.

Vizzini et al. [4] included, under the name *M. robertiana*, the sequence of a fungus collected in Italy and with macro- and microscopic agreement with the original description by Bon [28]. It is different from any other species of the *Urticocystis* Boekhout [2,3,4,5]. We decided to describe it as a new species here.

***Melanoleuca acystidiata*** Para, Antonín, Ševčíková, Ďuriška & Tomšovský, sp. nov. Figure 6 and Figure 7.

Mycobank MB 838744

*Diagnosis*. It differs from *M. microcephala* by the pileus and stipe color and the white context in the stipe base, from *M. paedida* by the pileus color and slightly larger basidiospores (*M. paedida* 6.5–8.5 × 3.9–5.5 μm), and always absent cheilocystidia; from *M. stridula* by the pileus color and larger basidiospores (*M. stridula* 6.5–8(9) × 5–6 µm); and from *M. fontenlae* by the pileus and stipe color, white context in the stipe base, and absent cheilocystidia; all mentioned species also differ phylogenetically.

*Holotype*. Italy, South Tyrol, Bolzano Distr., St. Jacob in Val di Vizze, 46°54′03″ N, 11°28′14″ E, 1500 m a.s.l., numerous specimens on a grassy clearing of a *Abies alba* wood, 23 August 2007 leg. R. Para and R. Fontenla (ANC M0205, GenBank/EMBL: JN616462).

*Etymology*. Acystidiata—a species lacking cystidia.

Pileus 20–50 mm broad, from convex to irregularly flattened, with less distinct or absent umbo; margin inflexed; smooth, opaque, or silky-looking surface; hairless; slightly sticky when wet; finely velvety under lens; brown to dark brown (S60 Y70–99 M50, S80 Y10–99 M40) and sometimes almost black; sometimes paler or with ochre tinge (S50 Y40–50 M10, S40 Y 99 M20–40, S50 Y 99 M20–40); sometimes with discolored areas. Lamellae close, L = c. 60–70, l = 1–2 in between each lamella, emarginate, straight to slightly rounded, pure white to whitish or pale gray, sometimes darkening in age. Stipe usually (slightly) longer than pileus width, 30–70 × 2–7 mm, cylindrical, sometimes broadened at the base, surface glabrous, longitudinally fibrillose, sometimes twisted, in young basidiomata whitish, then light brown to brown with ochre tinge (S40 Y40–60 M10, S40 Y 99 M20–30), sometimes with indistinct whitish floccules. Context exiguous in the pileus; fibrous and tenacious in the stipe; white, whitish to light brown in the cortex; unchanging, with light herbaceous or fungoid smell and mild taste.

Basidiospores [96/3/3] (6.7)7.2–10(11) × 5–7.2(7.5) µm, average 8.2 × 6.1 µm, E = 1.19–1.64, E = 1.36; subglobose; broadly ellipsoid; ovoid; with ± large, regular, or irregular warts; ridges absent or rare; amyloid. Basidia [6/1/1] 29–37 × 11.5–13 µm, four-spored, clavate or subfusoid. Basidioles 12–35 × 5.0–13 µm, clavate, subcylindrical, subfusoid. Cheilocystidia and pleurocystidia not observed. Marginal cells 20–35 × 5.0–9.0 µm, clavate, (sub)cylindrical, mostly irregular, sometimes septate, thin-walled. Pileipellis acutis to ixocutis, consisting of cylindrical, thin-walled, up to 8.0 µm wide hyphae, inamyloid, smooth or with scattered simple to subcoralloid lateral projections; terminal cells adpressed to erect, cylindrical, narrowly fusoid or clavate, thin-walled, up to 10 µm wide; fusoid cystidioid elements rarely present. Stipitipellis a cutis of cylindrical, parallel, slightly thick-walled, up to 6.0 µm wide hyphae, inamyloid. Caulocystidia 22–40 × 7.0–12.5 µm, rare to frequent, in groups or isolated, clavate or cylindrical, sometimes rostrate. Clamp connections absent in all tissues.

*Ecology*. On soil in grass in montane spruce (*Picea abies*), fir (*Abies alba*), and larch (*Larix decidua*) stands on acidic soil of high elevation (≈1500–1900 m a.s.l.).

*Distribution*. *Melanoleuca acystidiata* has been found only in three montane localities in the north of Italy and Switzerland.

*Additional specimens examined*. Italy: Piemonte, Cuneo Distr., Bosco delle Navette di Ormea, alt. ≈1600 m a.s.l., 5 October 2015 leg. R Para and R. Fontenla (ANC M0233). — Switzerland: Davos, Schatzalp, alt. ≈1900 m, 25 June 2009 leg. J. Borovička (BRNM 772203).

*Remarks*. *Melanoleuca acystidata* is characterized by a more or less dark brown pileus; whitish lamellae; a stipe shorter or longer than the pileus width that is whitish when young, then light brown; a white context; subglobose; broadly ellipsoid or ovoid basidiospores, 7.2–9 × 6–7.2 µm in size; the absence of any hymenial cystidia; a pileipellis in the form of an (ixo)cutis with adpressed to erect terminal cells; and one type of clavate or cylindrical caulocystidia.

From the phylogenetically related species, *M. microcephala* differs by a (grayish) brown, beige-gray, grayish or uniformly grayish brown pileus, a gray-brown to dark gray-brown stipe, and a dark (reddish) brown to black-brown context in the stipe base. *Melanoleuca paedida* differs by an ochraceous fawn to pale or dark gray-brown pileus, slightly smaller basidiospores, 6.5–8.5 × 3.9–5.5 μm, average 7.3 × 5.1 μm, and occasional cheilocystidia. *Melanoleuca stridula* has a uniformly rather dark, yellow-brown, (gray-)brown or dark brown pileus and smaller basidiospores, 6.5–8(9) × 5–6 µm, average 7.5 × 5.5 µm (our unpublished studies, [31,33,34]). *Melanoleuca fontenlae* has a whitish gray, gray-brown, dark brown pileus pallescent up to ochraceous brown or brown; an ochraceous yellowish to brownish, basally dark gray-brown stipe; a brown to dark brown context in the stipe base; and present cheilocystidia. The European collections of *M. angelesiana* have a larger, 25–80 mm broad, uniformly silvery gray, gray-brown, brown, dark brown, and centrally up to black-brown pileus and a more robust (45–80 × 4–10 mm), entirely floccose gray-brown stipe (our unpublished studies). The syntype of *M. angelesiana*, described from the Olympic Mts., Washington, USA, newly sequenced by us is more related to *M. acystidiata* Bon than to *M. angelesiana* identified by Vizzini et al. [4], but both species are clearly different. Among other species without cystidia, *Melanoleuca brachyspora* Harmaja differs by slightly smaller basidiospores, (6.5)7.0–9.0 × 4.5–6.5 μm, average = 7.5 × 5.5 μm. All above mentioned species also differ phylogenetically.

***Melanoleuca castaneofusca* Contu,** Bulletin de la Fédération Mycologiques Dauphiné-Savoie 150: 41, 1998. Figure 8.

*Holotype*. Italy: Sardinia, Orto Botanico di Cagliari, 3 November 1992 leg. M. Contu (CAG 921103-01!).

Pileus 24–100 mm broad, broadly conical to almost applanate, without umbo or with slightly distinct, broad, obtuse umbo at center, slightly reflexed towards margin, pileus cuticle exceeding the pileus margin, sometimes undulate, hygrophanous or not, smooth, glabrous or finely fibrillose, with silvery appearance in some pilei, not striate, uniformly dark gray-brown to brown (5YR3/1–2, 4/2; 7.5YR3/2–4, 4/1–2; 10YR4/1–4), pale brown (S30Y60M30) with darker brown center, without or with watery brownish stains. Lamellae moderately close, L = 40–60, l = 1–3(4), emarginate and adnexed with tooth, sinuate, rather narrow, cream-colored with a beige reflex (7.5YR8, 10YR8/1), edge concolorous, finely pubescent. Stipe usually shorter than pileus width, 12–70 × 4–10 mm, cylindrical, slightly broadened at apex, tapering towards the base, slightly clavate to clavate-bulbose (up to 15 mm) at base, distinctly floccose to floccose-tomentose at apex, otherwise longitudinally fibrillose, brownish to gray-brown or dark brown (4/2; 7.5YR3/2, 4/2; 10YR4/2–4), whitish or slightly brownish tinged at base; basal tomentum whitish. Context with slight fungoid smell and mild fungoid taste, white in pileus, brown beneath the pileipellis, fibrillose, whitish in stipe and with brownish yellow or pale orange tinge in the stipe base, in stipe cortex grayish.

Basidiospores [100/4/4] (6.0)6.5–8.0 × 4.0–6.0 μm, average = 7.1 × 4.9 μm, E = (1.20)1.28–1.70(1.75), Q = 1.47, (broadly) ellipsoid, ellipsoid-fusoid, with a suprahilar depression, with verruculose ornamentation, verruculae ± rounded, up to 0.5 μm high, variable in size, connections very rare to absent, amyloid. Basidia [28/4/4] 18–40 × 6.0–12 μm, four-spored, clavate to subfusoid. Basidioles 15–38 × 5.0–12 μm, clavate, subcylindrical, subfusoid. Cheilocystidia (length × base width × apex width) [31/3/3] 18–55 × 5.0–11 × 3.0–5.0 μm, urticoid, of the brevipes- and also exscissa-type, basal part fusoid, lageniform or clavate, septate or not, sometimes irregular, thin-walled, apical part subulate, obtuse with or without crystaliferous cap. Marginal cells 16–35 × 5.0–9.0 μm, fusoid, clavate, lageniform, irregular to (sub)coralloid, thin-walled. Pleurocystidia often absent, sometimes scattered, 30–40 × 5–10 μm, of the brevipes-type, with or without septum and apical incrustation. Trama hyphae cylindrical to inflated, thin-walled, inamyloid, up to 15 μm wide. Pileipellis an ixocutis, composed of cylindrical, ± thin-walled, gelatinized, up to 8.0 μm wide, smooth hyphae; terminal cells adpressed to (sub)erect, cylindrical, narrowly clavate, obtuse, thin-walled, with gray-vacuolar pigmentation. Stipitipellis a cutis of cylindrical, parallel, slightly thick-walled, up to 6.0 μm wide hyphae with gray-brown pigmentation. Caulocystidia (caulohymenium) of two types, (1) urticoid cystidia of the exscissa- or brevipes-type, 25–60 × 4.0–9.0 μm, both septate and not septate, thin-walled, with indistinct to distinct apical incrustation, and (2) clavate, subfusoid, or subcylindrical cells, 18–50 × 7.0–12 μm, thin-walled; cystidia of the type (1) may be very rare or even absent. Clamp connections absent.

*Ecology.* On soil under *Ulmus* (BRNM 761900), in a greenhouse on a bare soil (BRNM 761901), on sandy soil under *Picea* (LIP RC86021), in a flowerpot with *Mentha* sp. in an urban apartment (SLO 1639), on composted soil in a cemetery [35], and on soil and shredded bark mulch in a botanic garden (K(M) 92562); holotype on basic sandy soil near *Cactaceae* in the botanic garden (CAG 921103-01). The species may be associated with commercial soil and compost substrates.

*Distribution*. This species is now confirmed from the Czech Republic, France, Great Britain (England), Italy, and Slovakia until now. However, we suppose it is more broadly distributed in similar habitats.

*Specimens examined.* Czech Republic: Osek near Hořovice, Vystrkov, 12 November 2012 leg. O. Jindřich (BRNM 761901).—France: Landrellec (Cotes du Nord), près Pleumeur-Boudon, 21 December 1986 leg. Reaudin, det. R. Courtecuisse (LIP RC86021, as *M. brevipes*).—Great Britain: England, Surrey, Kew, Royal Botanic Gardens, 9 October 2001 leg. A. Henrici (K(M) 92562, as *M. turrita*, a mixed collection with *M. bataillei*).—Italy: Sardinia, Orto Botanico di Cagliari, 3 November 1992 leg. M. Contu (CAG 921103-01, holotype).—Ravenna, Pineta di San Vitale, Casa vecchia, 8 November 2007 leg. M. Calderoni (BRNM 761900).—Slovakia: Podunajská nížina lowland, Bratislava, 28 October 2013 (SLO 1639).

*Remarks. Melanoleuca castaneofusca*, described from Sardinia [36], is characterized by rather robust basidiomata with a dark-colored pileus; cream lamellae; an apically distinctly floccose to floccose-tomentose, brownish to gray-brown stipe; whitish and brownish yellow tinged context in the stipe base; rather small basidiospores; urticoid cheilocystidia of the brevipes- and exscissa-type; a pileipellis in the form of an ixocutis; and well-developed caulohymenium. This species produces its basidiomata in late autumn in habitats strongly influenced by humans or at completely artificial places (e.g., greenhouse, cemetery, botanic garden). In recent literature, only one collection (Great Britain, Surrey, Morden Cemetery, October 2012 leg. T. Brown) was published [35]; it was identified on the basis of our sequences.

Among phylogenetically similar species, *M. luteolosperma* differs by smaller basidiomata (pileus 22–35 mm, stipe 50–55 × 3–4 mm), an ochraceous yellow or pale gray-brown stipe, and slightly larger basidiospores [7.0–9.0 × 5.0–5.5(6.0) μm]. *Melanoleuca paedida* differs by a smaller, 40–50 mm broad, ochraceous fawn to pale or dark gray-brown pileus and lacks a caulohymenium. *Melanoleuca fontenlae* differs by smaller basidiomata (pileus 9–39 mm broad, stipe 17–60 mm long), only finely floccose-pubescent stipe, a brown to dark brown context in the stipe base, and larger basidiospores [(6.0)6.5–10 × (4.0)4.8–6.0(6.5) μm, average = 8.0 × 5.6 μm].

*Melanoleuca malenconii* Bon is macroscopically somewhat similar but differs by pale to grayish yellow lamellae when mature; a slightly pruinose-pubescent to distinctly floccose, sometimes especially in lower part floccose-hairy stipe that is rarely subglabrous at apex; the presence of pleurocystidia; longer basidiospores (up to 10 μm); and sometimes also by relatively smaller, paler basidiomata [2,28]. *Melanoleuca humilis* (Pers.) Pat. is morphologically similar species different in its grayish or beige lamellae in mature basidiomata, a (dark) brown context in the stipe base, longer basidiospores (up to 10 μm), and often also a smaller pileus [2].

***Melanoleuca luteolosperma*** (Britzelm.) Singer, Cavanillesia 7: 127, 1935. Figure 9, Figure 10 and Figure 11.

≡ *Agaricus luteolospermus* Britzelm., Bericht der Naturhist Vereins in Augsburg 31: 160, 1894. ‒ *Tricholoma luteolospermum* (Britzelm.) Lapl., Dictionnaire Iconographique des Champignons Supérieurs: 540, 1894.

*Lectotype*. Britzelmayr, Bericht der Naturhistorischen Vereins in Augsburg 31: Table 647, 1894 [37].

*Epitype*. Germany, vicinity of Augsburg, Wallenburg, Bergheim, Siebentischwald, Stadtberg n. Stätzling, Eurasburg, Peterhof, Wittelsbacher Park, 10 June 1962 leg. J. Stangl (M 0139486! [37]).

= *Melanoleuca pseudoluscina* Bon, Documents Mycologiques 10(37–38): 89, 1979.

*Holotype*. France, Somme, Quend les Pins, fourrés de l’*Hyppophaeion* et *Koelerion*, October 1963 leg. M. Bon (LIP 31018!).

*Epitype.* Italy, Venezia, Pineta Ca’ Savio di Cavallino-Treporti, 12 December 1999 leg. E. Bizio (ANC M0192).

= *Melanoleuca pseudopaedida* Bon, Documents Mycologiques 20(79): 58, 1990.

*Holotype*. France, Quend les Pins (Somme), Panne de dunes, *Salix* et *Pinus*, 5 November 1983 leg. M. Bon (LIP, Bon 831105!).

Pileus 22–42(62) mm broad, convex with obtuse center, almost without umbo or applanate, depressed at center, inflexed to straight at margin, smooth or finely pruinose, glabrous, never translucently striate, uniformly pale ochraceous gray to gray (5A2–5B3, 5C1–5D1), grayish brown (6D–E4) or brown (6D7) with whitish very margin. Lamellae moderately close, L = c. 40–60, l = 2–3, emarginate and attached with a tooth, ventricose, broad, white to pale cream-colored (2–3A2), with concolorous, uneven, finely pubescent edge. Stipe shorter or longer than pileus width, 20–55 × 3–7 mm, cylindrical, slightly broadened at apex, slightly broadened or attenuated and shortly radicating at base, finely longitudinally fibrillose, finely floccose-pubescent at apex, whitish at apex, pale ochraceous yellow (3A2) or pale gray-brown (±6C3) otherwise. Context whitish in pileus, whitish in stipe apex and (dark) brown (±6E4) in base, with indistinct smell and mild to bitterish taste.

Basidiospores [180/8/8] (6.0)7.0–9.0 × (4.0)4.2–5.5(6.5) μm, average 7.7 × 5.1 μm, E = (1.20)1.26–1.74(1.8), Q = 1.50, broadly ellipsoid, broadly ellipsoid, ellipsoid-fusoid, subovoid, verruculose, warts mostly irregular in shape and variable in size, up to 0.75 μm high, amyloid, ridges rare. Basidia [38/6/6] 26–46 × 8.0–12 μm, four-spored, clavate, subfusoid. Basidioles (12) 17–40 × 4.0–12 μm, clavate, subfusoid, cylindrical. Cheilocystidia [80/8/8] (length × base width × apex width) 22–66 × 3.0–10 × 3.0–4.5 μm, urticoid, of the exscissa- and brevipes-type, basal part clavate, subvesiculose, subfusoid, sometimes irregular, apex subulate or (sub)cylindrical, thin- to slightly thick-walled, muricate or not. Marginal cells 12–24 × (4.0)8.0–12 μm, clavate, vesiculose, sometimes two- or three-celled, thin-walled. Pleurocystidia absent. Trama hyphae cylindrical or subinflated, thin-walled, inamyloid, 3.0–15 μm wide. Pileipellis an (ixo)cutis, sometimes transient to subtrichoderm, composed of radially arranged, cylindrical, thin-walled, inamyloid, 3.0–8.0 μm wide hyphae; terminal cells often cystidioloid, adpressed to erect, cylindrical to clavate or fusoid, sometimes irregular to branched, thin-walled, obtuse, up to 8.0 μm wide, grayish brown in KOH. Stipitipellis a cutis of cylindrical, parallel, slightly thick-walled, inamyloid, 3.0–6.0 μm wide hyphae. Caulohymenium of (1) 16–70 × 2.0–11 μm, clavate, cylindrical, thin-walled, sometimes uniseptate cells; (2) urticoid cystidia, 24–64 × 4.0–10 μm, similar to cheilocystidia; and (3) caulobasidia, four-spored. Clamp connections absent.

*Ecology*. On soil, under *Picea abies* and *Fagus sylvatica* in a montane forest and on the riverbank under *Populus*, *Sambucus*, and *Alnus* (Slovakia), on the riverbank among grass, *Dryas*, and *Picea* (Slovenia, along the path in decaying remnants of grasses; under *Juglans regia, Rosa*, and *Clematis* (Italy); sandy soil under *Populus* (Italy); under *Cistus* (Italy); in a thermophilic forest stand (Czech Republic); and in a semi-dry grassland with *Pinus* (Germany).

*Distribution*. This species is widely distributed in Europe. Its occurrence is confirmed from the Czech Republic, Germany, Italy, Slovakia, and Slovenia.

*Specimens examined*. Czech Republic: Bohemian Karst, Koda, 28 October 2009 leg. J. Burel and O. Jindřich (BRNM 817820).—France: Somme, Quend les Pins, thickets of *Hyppophaeion* and *Koelerion*, October 1963 leg. M. Bon (LIP 31018, holotype of *M. pseudoluscina*).—Quend les Pins (Somme), dunes, *Salix* and *Pinus*, 5 November 1983 leg. M. Bon (LIP, Bon 831105, holotype of *M. pseudopaedida*).—Germany: vicinity of Augsburg, Wallenburg, Bergheim, Siebentischwald, Stadtberg n. Stätzling, Eurasburg, Peterhof, Wittelsbacher Park, 10 June 1962 leg. J. Stangl (M 0139486, epitype).—Lahr-Sulz, *Pinus*, 20 June 1998 leg. G. Saar (BRNM 772190).—Italy: Calabria, Pollino National Park, Lungro, Piano di Campolungo, 8 October 2019 leg. H. Ševčíková (BRNM 826043).—Monti Sibillini,

Altino, 22 October 2010 leg. V. Antonín (BRNM 817785).—Bibione, Faro, 8 December 2014 leg. E. Campo (817784).—Venezia, Pineta Ca’ Savio di Cavallino-Treporti, 12 December 1999 leg. E. Bizio (ANC M0192, epitype of *M. pseudoluscina*).—Ferrara, loc. Bosco Spada di Codigoro (FE), 14 November 2008 leg. G. Consiglio and A. Gennari (ANC M0194, as *M. pseudoluscina*).—Venezia, Giardini Ca’ Bianca, 14 November 1993 leg. E. Bizio (ANC M0191, as *M. pseudoluscina*).—Venezia, loc. Pineta Ca’ Savio di Cavallino-Treporti, 20 December 1998 leg. L. Levorato and C. Losi (ANC M193, as *M. pseudoluscina*).—Belluno, Caviola, Bosco Di Biasio di Falcade, 16 August 1996 leg. E. Bizio (ANC M0195, as *M. pseudoluscina*). — Slovakia: Velká Fatra Mts., Lubochňa, Lubochňanská dolina, Kundračka National Nature Reserve, alt. 820–1280 m a.s.l., 31 August 2002 leg. V. Kabát and V. Antonín (BRNM 761907).—Považský Inovec, Moravany nad Váhom, Výtoky, 29 September 2012 leg. I. Stach (SLO 1591).—Biele Karpaty, Skalka nad Váhom, near river Súčanka, 10 October 2013 leg. O. Ďuriška (SLO 1632).—Devínska Kobyla, Fialková dolina valley, 28 November 2019 leg. O. Ďuriška (SLO 2515).—Slovenia: Julians Alps, Triglav National Park, Zadnja Trenta, Upper Soča valley, alt. 980 m a.s.l., 10 October 2001 leg. G. Podgornik (BRNM 772201).

*Remarks. Melanoleuca luteolosperma* is an extremely variable species in terms of colors, characterized by rather small basidiomata with a uniformly pale ochraceous gray to gray or grayish brown pileus; pale cream lamellae; a cylindrical, basally slightly broadened or attenuated stipe that is finely floccose-pubescent only at apex and pale ochraceous yellow or pale gray brown; a pale brown context; moderately large basidiospores ((6.0)7.0–9.0 × (4.0)4.2–5.5(6.5) μm), varying from broadly ellipsoid (Q = 1.38) to ellipsoid (Q = 1.59); urticoid cheilocystidia of both the exscissa- and brevipes-type; a pileipellis often forming cystidioloid terminal cells; and mostly well-developed caulohymenium. The collection from Italy (Altino, BRNM 817785) differs by smaller basidiospores [6.5–7.5 × 4.5–5.5(6.0) μm].

Métrod [33] and Bon [10] considered *M. luteolosperma* a species with macrocystidia. However, Fontenla and Para [37] selected the epitype from the material collected in the original locality [38]. This epitype has urticoid cheilocystidia. *M. luteolosperma* was described as having a yellowish white spore print and whitish to yellowish basidiospores; its lamellae should be white when young, or whitish or brownish white [39]. The sequence of this epitype is identical to several recent collections with white, whitish, or only slightly cream lamellae. Therefore, we conclude that *M. luteolosperma* is a fungus with a broader variability of the lamellae color and with urticoid cystidia.

The sequence identified as *M. pseudoluscina* (JN616455) from Italy [4] was grouped among those of *M. luteolosperma*. Unfortunately, the sequencing of the holotype specimen of *M. pseudoluscina* failed. The holotype specimen (LIP 31018) is described as having a gray, often initially pale, then gray-brown or fuligineous and finaly black-gray (“atro-ardosiacus”) pileus, and a brown (“aerino-fuscus”) or fuligineo-gray (“fuligonoso-ardosiacus”) stipe [40]; microscopically, except for slightly different basidiospores [7.0–9.0(9.5) × 5.2–6.0(6.5) μm, average 8.1 × 5.7 μm], it agrees with the description of *M. luteolosperma* above. The holotype specimen of *M. pseudopaedida* is phylogenetically identical with *M. luteolosperma*, but different from *M. pseudopaedida* sensu Vizzini et al. [3], which is described as a new species *M. fontenlae* above.

Among phylogenetically close species, *Melanoleuca castaneofusca* differs by larger basidiomata, a darker dark gray-brown to brown pileus, a darker brownish to gray-brown or dark brown stipe, and slightly smaller basidiospores (6.5–8.0 × 4.0–6.0 μm). *Melanoleuca paedida* differs by a slightly larger, 30–60 mm broad, ochraceous fawn to pale or dark gray-brown pileus; a stipe that is brownish, ochraceous orange, or concolorous with the pileus; slightly different basidiospores; and a lack of a caulohymenium. *Melanoleuca fontenlae* differs by a darker pileus and stipe and larger basidiospores [(6.0)6.5–10 × (4.0)4.8–6.0(6.5) μm, average = 8.0 × 5.6 μm].

The European collections of *M. angelesiana* A.H. Sm. are macroscopically similar to *M. luteolosperma*. It differs by larger basidiomata (pileus 25–80 mm, stipe 45–80 × 4–10 mm); a whitish context in the stipe base; larger basidiospores, 7.0–10 × (4.5)5.0–6.5(7.0) µm; and large basidia and basidioles (30–55 × 10–12 μm) (our unpublished studies). However, this species was originally described in North America, and the identity of European collections in terms of the work of Vizzini et al. [3] and American specimens is questionable.

***Melanoleuca microcephala*** (P. Karst.) Singer, Cavanillesia 7: 123, 1935. Figure 12.

≡ *Tricholoma microcephalum* P. Karst., Hedwigia 20 (12): 177, 1881.

*Lectotype.* Finland, Tavastia australis, Tammela, Mustiala, 26 August 1881 leg. et det. P.A. Karsten (H; Herbarium Petter Adolf Karsten, no. 1604, as *Tricholoma microcephalus*, designated here, MycoBank MBT 395959).

Pileus 18–60 mm broad; broadly low convex-conical; conical to applanate and centrally depressed with low, broad, and less distinct umbo; inflexed to involute at margin, with pileipellis slightly projecting beyond the margin; hygrophanous; sometimes slightly translucently striate at very margin; smooth, glabrous, or slightly aeriferous at margin; finger test + or 0; (grayish) brown (7–8F3–6, 6E–F4) at center; beige-gray or grayish (6B–E2–3, 6D3–4), otherwise uniformly grayish brown (6C–D–E4–5). Lamellae moderately close, L = c. 30–60, l = 2–4, emarginate and attached to shortly decurrent with tooth, sinuate when young, up to ≈5 mm broad, whitish to pale cream-colored with beige reflex, with concolorous, finely pubescent edge. Stipe usually longer than pileus width, 22–105 × 2–5.5 mm; cylindrical; clavate to subbulbose towards base (9–12 mm); slightly broadened at apex; finely pruinose or floccose, especially at apex; longitudinally fibrillose or almost fibrillose-squamulose; whitish to pale brownish at apex; pale yellowish with ochraceous tinge (near 4A3); gray-brown (6F2–5) to dark gray-brown towards base; fibrils often darker than ground color when moist—therefore, the stipe gives a mottled appearance; basal tomentum white. Context whitish in pileus, brownish under the pileipellis, fibrillose and pale to dark (reddish)brown (6–7F6) to black-brown in stipe base, brownish in stipe cortex, without smell or with a slight earthy smell and mild taste.

Basidiospores 7.0–9.5(10) × 5.0–7.0 μm, average = 8.3 × 5.8 μm, E = 1.21–1.67, Q = 1.42, broadly ellipsoid, ellipsoid, ellipsoid-fusoid, obovoid, thin- to slightly thick-walled, ornamentation verrucose, sometimes with rare ridges, warts variable in shape and size, round or irregular, close to moderately close, amyloid. Basidia 28–37 × 9.0–13 μm; four-, very rarely two-spored; clavate. Basidioles 12–35 × 5.0–13 μm, clavate, subcylindrical, subfusoid. Cheilo- and pleurocystidia observed. Marginal cells 17–28 × (3.5)5.0–11 μm, cylindrical, clavate, subfusoid, utriform, often irregular, thin-walled. Trama hyphae cylindrical to subinflated, thin-walled, inamyloid, 3.0–15 μm wide. Pileipellis an ixocutis (margin) to ixosubtrichoderm (centre), composed of radially arranged, ± cylindrical, thin-walled, up to 10 μm wide hyphae with often grayish to gray-brown pigmentation; terminal cells adpressed to (sub)erect, up to ≈50 × 3.0–10 μm, clavate, cylindrical, subfusoid, obtuse, thin-walled; pale grayish yellowish in KOH. Stipitipellis a cutis of cylindrical, parallel, ± slightly thick-walled, 3.0–6.0 μm wide hyphae. Caulocystidia in groups, 18–62 × 5.0–15 μm, cylindrical, clavate, subfusoid, thin-walled. Clamp connections absent.

*Ecology*. On soil in the montane and alpine belt, on a pasture with scattered *Picea*, in an alpine meadow, in mosses and vegetation along the montane stream, in grass in spruce forest, and along the forest path. It seems to be a montane calciphilous species.

*Distribution*. It is known from the montane and alpine belt on altitudes of ≈1500 m or more a.s.l. It is only confirmed from Finland (lectotype), Italy, and Slovakia until now.

*Additional specimens examined***.** Italy: South Tyrol, Schlern-Rosengarten Naturpark (Sciliar-Catinaccio Nature Park), Tiers (Tires), alt. 1647 m a.s.l., 4 September 2013 leg. V. Antonín 13.212 (BRNM 817786). — South Tyrol, Prato Nuovo (Neuwies), Franzenshöhe, N 46°31′55″, E 010°28′46″, alt. 2150 m a.s.l., 30 July 2018 leg. B. Dima (BRNM 809300). — South Tyrol, Sulda (Solden), N 46°31′24″, E 010°35′17″, alt. 1864 m a.s.l., 31 July 2018 leg. J. Hrabáková (BRNM 817789). — South Tyrol, Franadega di Dobbiaco (BZ), alt. 1600 m a.s.l., 21 July 2005 leg. W. Tommasi (ANC M0196/JN616449 and herb. R. Para 050721-02, originally as *M. stepposa*). — Slovakia: Velká Fatra Mts., Ružomberok, Podsuchá part, Smrekovica, Skalná Alpa National Nature Reserve, alt. 1290–1320 m a.s.l., 23 September 2009 leg. V. Antonín (BRNM 817787 and 761890). — Velká Fatra Mts., Ružomberok, Podsuchá part, Smrekovica, protecting belt of the Skalná Alpa National Nature Reserve, alt. 1300–1367 m a.s.l., 24 September 2009 leg. V. Antonín (BRNM 817788). — Belianske Tatry Mts., Tatranská Kotlina, Skalné vráta, alt. ≈1450–1950 m a.s.l., 5 September 2001 leg. I. Milan (BRNM 772196).

*Remarks*. *Melanoleuca microcephala* is characterized by moderately large basidiomata with a (grayish) brown, beige-gray, grayish, or uniformly grayish brown pileus; whitish to pale cream lamellae; a gray-brown to dark gray-brown stipe; a dark (reddish)brown to black-brown context in the stipe base and the absence of cheilo- and pleurocystidia and the absence of caulohymenium. Two cystidioid elements, 35–36 × 7.0–8.0 μm, on the lamellar edge were found only in the Italian collection (Italy, Tiers, BRNM 817786). However, they were not typically urticoid cells, but structures such as a transient form between cystidia and marginal cells. An atypical pale orangish yellow coloring of the stipe after touching appeared in the Slovak collection (Ružomberok, BRNM 817788). Caroti et al. [41] published a collection of *M. microcephala* with a small, 15–20 mm broad, dark brown pileus with a white outermost margin and scattered urticoid cheilo- and pleurocystidia. However, this collection was not confirmed by phylogenetic studies and its identity is unclear.

In comparison with phylogenetically similar species, *Melanoleuca paedida* differs by an ochraceous fawn to pale or dark gray-brown pileus, a stipe of the same length or shorter than the pileus diameter, a slightly darker context in the stipe base, and slightly smaller basidiospores; the collection with cystidia as well as the acystidiate collection are known in this species. *Melanoleuca stridula* has a uniformly rather dark, yellow-brown, (gray-)brown, or dark brown pileus and smaller basidiospores, 6.5–8(9) × 5–6 µm, average 7.5 × 5.5 µm (our unpublished studies, [31,33,34]). *Melanoleuca acystidiata* has a dark brown pileus; a whitish, then light brown stipe; and a dark brown context in the stipe base. Among other species without cystidia, *M. brachyspora* Harmaja differs by a differently colored, brownish gray or brown to dark brown pileus; a white context in the stipe base; and slightly smaller basidiospores, (6.5)7.0–9.0 × 4.5–6.5 μm, average = 7.5 × 5.5 μm. The European collections of *M. angelesiana* differs by a whitish context in the stipe base and large basidia and basidioles (30–55 × 10–12 μm) (our unpublished studies, see also above).

***Melanoleuca paedida* (Fr.) Kühner & Maire**, Bulletin trimestriel de la Société mycologique de France 50: 18, 1934. Figure 13.

≡ *Agaricus paedidus* Fr., Epicrisis Systematis Mycologici: 53, 1838. ‒ *Gyrophila paedida* (Fr.) Quél., Enchiridion Fungorum: 18, 1886. ‒ *Tricholoma paedidum* (Fr.) Quél., Mémoires de la Société d’Émulation de Montbéliard, Série 2, 5: 341, 1873.

*Neotype*. Fries, Icones selectae Hymenomycetum nondum delineatorum, Table 46, Figure 1, 1867–1884 (designated here, MycoBank MBT 395960).

Pileus 30–60 mm broad, convex, soon flattened, then with depressed center, with absent or very reduced umbo and entire margin, glabrous (silky under lens, positive finger test), dry, opaque, ochraceous fawn to pale or dark gray-brown, with darker center. Lamellae moderately close to close, with numerous lamellulae of various lengths, emarginate and attached with tooth, high, straight, whitish or brownish gray, with concolorous edge. Stipe shorter than 30 × 6 mm or 60 × 4–8 mm, shorter or of the same length as the pileus width, cylindrical, enlarged or clavate at the base, with barely pruinose surface at the apex only, smooth or striate, brownish, ochraceous orange or concolorous with the pileus, with widespread white mycelial felt at the base. Context white or whitish, immutable, slightly darker in the stipe base, brownish also in the cortex of the stem and between the pileus and the lamellae, with pleasant herbaceous smell.

Basidiospores 6.5–8.5 × 3.9–5.5 μm, average 7.3 × 5.1 μm, E = 1.20–1.76, Q = 1.43, ellipsoid or subglobose, with small to medium sized, not close, amyloid warts. Basidia 26–40 × 9–10 μm, clavate, with long and narrow base, four- and two-spored. Cheilocystidia 50–74 × 5–10 μm, very numerous, urticoid. Pleurocystidia numerous, similar to cheilocystidia. Cheilo- and pleurocystidia may sometimes absent. Marginal cells 20–55 × 5–10 μm, scattered, clavate, twisted. Trama hyphae parallel, cylindrical. Pileipellis an ixocutis of interwoven hyphae, 3–4 µm wide, with a very thin gelatinous layer, with lemon-yellow intracellular pigment in superficial hyphae, yellowish-brown in underlying hyphae. Stipitipellis of parallel hyphae, with rare emerging hairs of variable shape. Caulocystidia absent. Clamp connections absent.

*Ecology*. On soil in *Cedrus atlantica*, *Cedrus brevifolia*, and *Cedrus libani* litter (Calabria) and in the grass under *Pinus* sp. (Lombardia).

*Distribution*. Recently confirmed only from Italy, but its distribution may be broader in Europe.

*Specimens examined*. Italy: Calabria, Colamauci di Celico (CS), alt. 1200 m a.s.l., 16 November 2009 leg. C. Lavorato (ANC M0189, JN616452). — Lombardia, lago Cancano di Valfurva (SO), alt. 1900 m a.s.l., 7 September 2001 leg. E. Carassai (herb. R. Para 010907-02).

*Remarks*. *Melanoleuca paedida* is characterized by an ochraceous fawn to pale or dark gray-brown pileus with darker center; whitish or brownish gray lamellae; a stipe shorter or of the same length as the pileus diameter; brownish, ochraceous orange, or concolorous with the pileus; a slightly darker context in the stipe base, moderately large, ellipsoid, or subglobose basidiospores; and lacking caulocystidia. Cheilo- and pleurocystidia are either present (ANC M0189) or absent (010907-02). The variability of the latter character requires further study. Therefore, the epitype is not proposed here. The identity of this species is based on the studies by Vizzini et al. [4].

Among phylogenetically similar species, *Melanoleuca castaneofusca* differs by a larger, uniformly dark gray-brown to brown pileus with watery brownish stains, brownish to gray-brown or dark brown stipe that is a distinctly floccose to floccose-tomentose at apex, constantly present cheilocystidia, and well-developed caulohymenium. *Melanoleuca luteolosperma* has a smaller, 22–35 mm broad pileus with a whitish very margin, a pale ochraceous yellow or pale gray-brown stipe, slightly smaller basidiospores, constantly present cheilocystidia, and a well-developed caulohymenium. *Melanoleuca fontenlae* differs by a brown to dark brown context in the stipe base and larger basidiospores [(6.0)6.5–10 × (4.0)4.8–6.0(6.5) μm, average = 8.0 × 5.6 μm], and it usually has a well-developed caulohymenium or at least caulocystidia. *Melanoleuca stridula* has a longer stipe than the pileus dimeter, slightly lager basidiospores [7.0–8.5(9.0) × 4.7–6.0 μm, average = 7.9 × 5.4 μm], constantly absent cheilocystidia, and a developed caulohymenium (our unpublished studies). *Melanoleuca acystidiata* differs by a dark brown pileus; a whitish, then light brown stipe; a white context in the stipe base; slightly larger basidiospores, 7.2–9 × 6–7.2 µm, average 8.6 × 6.3 µm; and always absent cheilocystidia.

European collections of *M. angelesiana* are also morphologically similar. They differ by a larger, 25–80 mm broad, uniformly silvery gray, gray-brown, brown, dark brown, and centrally up to black-brown pileus; a more robust (45–80 × 4–10 mm), gray-brown stipe; larger basidiospores [7.0–10 × (4.5)5.0–6.5(7.0) μm, average = 8.5 × 5.8 μm]; and consistently absent cheilocystidia and clavate caulocystidia.

## 4. Discussion

*Melanoleuca* is an example of an agaricomycete genus where molecular revision of type specimens is crucial for species concept and species recognition [42]. The several European *Melanoleuca* species proposed by Bon (e.g., [10,28]) were later synonymized [2–4] due to results of DNA sequence analyses. Although taxonomy of *Melanoleuca* in Europe has a long tradition, a new can be discovered here, e.g. *Melanoleuca juliannae* Rimóczi et al., was described recently [5]. Both recently described species *M. galbuserae* and *M. juliannae* share a preference for the specific grassland habitats (alpine grasslands or Pannonian sand grasslands), although their habitat preferences may be broader. Ecological differences can be found also among other species. *Melanoleuca fontenlae* prefers sandy soil and dunes, *M. acystidiata* and *M. microcephala* seem to be restricted to montane/alpine habitats, *M. paedida* prefers soils with coniferous litter, and *M. luteolosperma* occurs in a wide range of habitats. *M. castaneofusca* has been found in many habitats affected by human activity and may be dispersed by commercial substrates. 

*Melanoleuca* is an extremely difficult genus for species identification due to overlapping morphological characters; the most useful distinguishing characters are noted here. The context color at the stipe base is the most important macroscopic characters in *Melanoleuca*—in the *M. castaneofusca* group, *M. acystidiata* and *M. paedida* have a white context, while *M. fontenlae*, *M. microcephala*, and *M. luteolosperma* differ by a darker context; *M. castaneofusca* has a whitish context with a brownish yellow or pale orange tinge. In the *M. exscissa* group, *Melanoleuca galbuserae* has a white to only slightly grayish context in the stipe base, which is similar to *Melanoleuca exscissa* and *Melanoleuca rasilis*; other European species (*Melanoleuca diverticulata*, *Melanoleuca stepposa* Vacek, and *Melanoleuca tristis*) have a dark-colored stipe base context [3]. The white stipe base context is also known in *M. porphyropoda* described from China, but has also been reported in the UK on the basis of their morphological features [16]. *Melanoleuca galbuserae* and *M. porphyropoda* seem to be related according to ITS data (Figure 1).

Moreover, caulohymenium [11] is a useful distinguishing character in *Melanoleuca*—*M. stepposa* and *M. tristis* differ from *M. galbuserae* and *M. griseobrunnea* by the well-developed caulohymenium [3]. Such caulohymenium can be found also in *M. castaneofusca*, *M. fontenlae*, and *M. luteolosperma*.

*Melanoleuca castaneofusca* can be well distinguished also by a distinctly floccose to floccose-tomentose stipe apex and a pale-colored stipe; other related species (*M. luteolosperma*, *M. paedida*, *M. fontenlae*) have a darker-colored stipe with only a finely pruinose to pruinose-flocculose apex.

The basidiospore dimensions are crucial for identification in many agaricomycetes, but in *Melanoleuca*, this character may be confusing, because both, four- and two-spored basidia can occur in one hymenium and respective spores differ in their dimensions [43]. *Melanoleuca paedida* has both two- and four-spored basidia in the hymenium; only four-spored or only rare two-spored basidia were found in other species. Two species, *M. microcephala* and *M. acystidiata*, consistently lack cheilocystidia, but both cystidiate and acystidiate collections are known in *M. paedida*.

## Figures and Tables

**Figure 1 jof-07-00191-f001:**
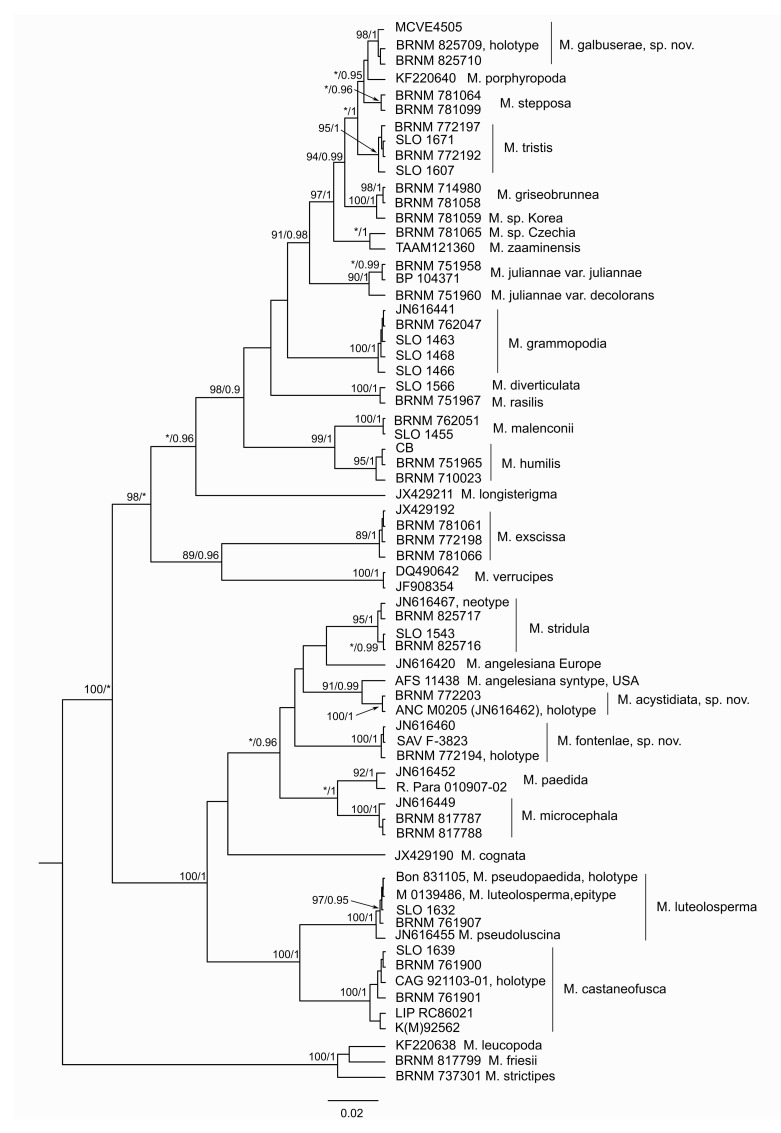
Phylogenetic tree of the ITS region of *Melanoleuca* species conducted by Bayesian analysis in BEAST 2 (for legends to numbers, see Table 1). Numbers at branches indicate maximum likelihood bootstrap proportion and Bayesian posterior probability values. The asterisks (*) mark low support (<75 in maximum likelihood; <90 in Bayesian analysis). The bar indicates the number of expected substitutions per site.

**Figure 2 jof-07-00191-f002:**
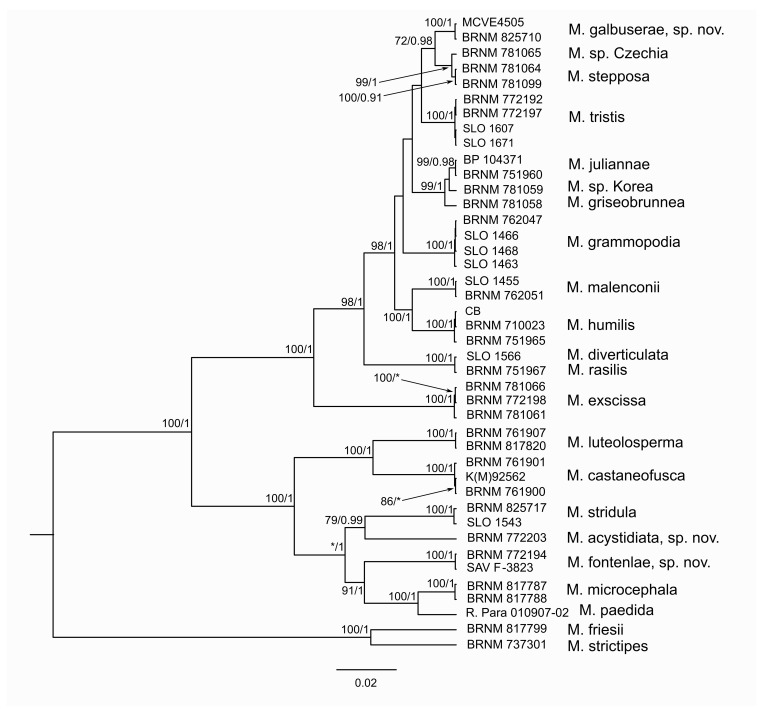
The species tree of ITS–tef1–rpb2 genes of *Melanoleuca* species conducted by multispecies coalescent analysis (for the legend to numbers, see Table 1). Numbers at branches indicate maximum likelihood bootstrap proportion and Bayesian posterior probability values. The asterisks (*) mark low support (<75 in maximum likelihood; <90 in Bayesian analysis). The bar indicates the number of expected substitutions per site.

**Figure 3 jof-07-00191-f003:**
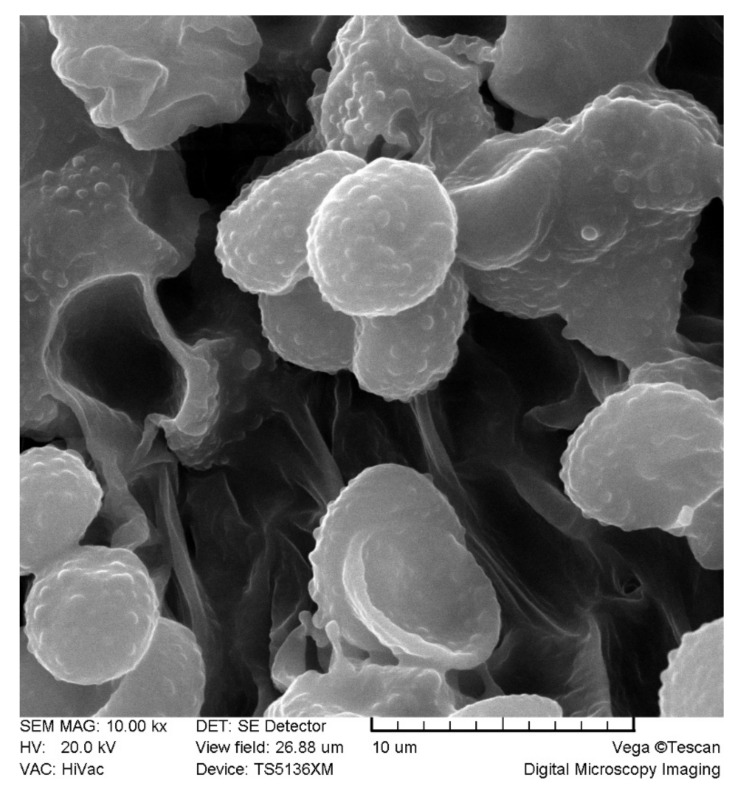
*Melanoleuca galbuserae* (holotype). SEM microphotographs of basidiospores. Photo L. Ilkovics.

**Figure 4 jof-07-00191-f004:**
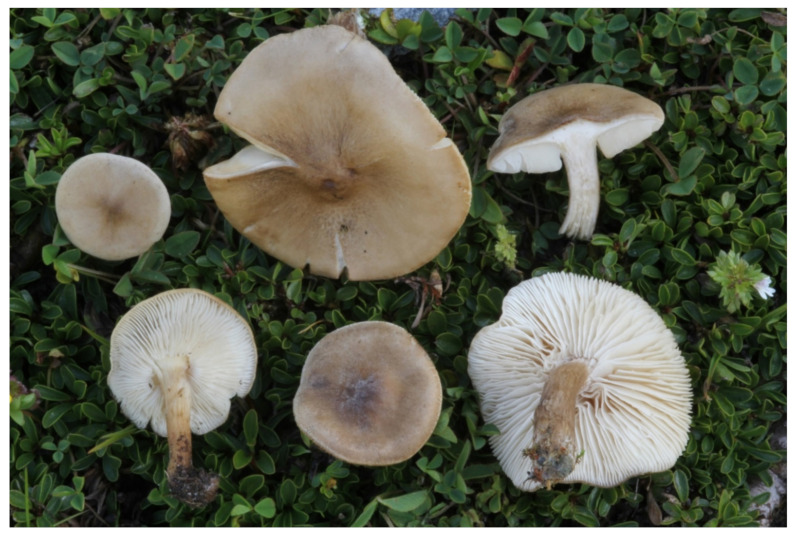
*Melanoleuca galbuserae* (holotype). Italy, Solda, Refugio Milano, 29 July 2018, photo H. Ševčíková.

**Figure 5 jof-07-00191-f005:**
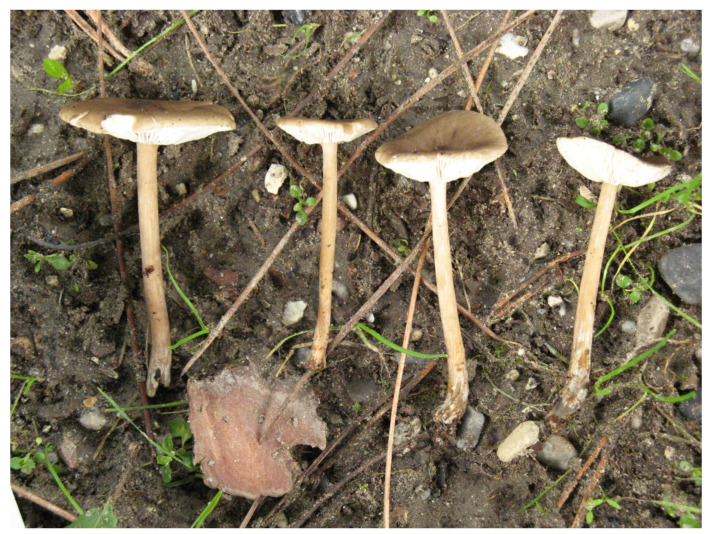
*Melanoleuca fontenlae* (BRNM 825714). Italy, Ravenna, Pineta Ramazzotti and Dunes di Lido di Dante, 5 November 2007, photo V. Antonín.

**Figure 6 jof-07-00191-f006:**
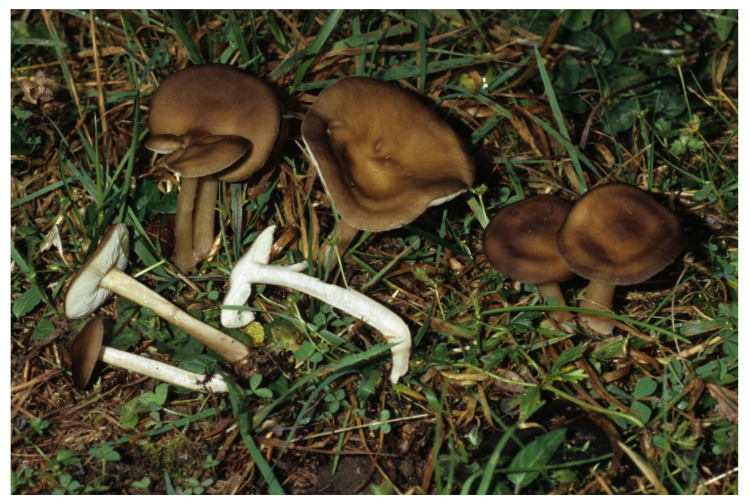
*Melanoleuca acystidiata* (holotype). Italy, Bolzano Distr., St. Jacob in Val di Vizze, 23 August 2007, photo R. Para.

**Figure 7 jof-07-00191-f007:**
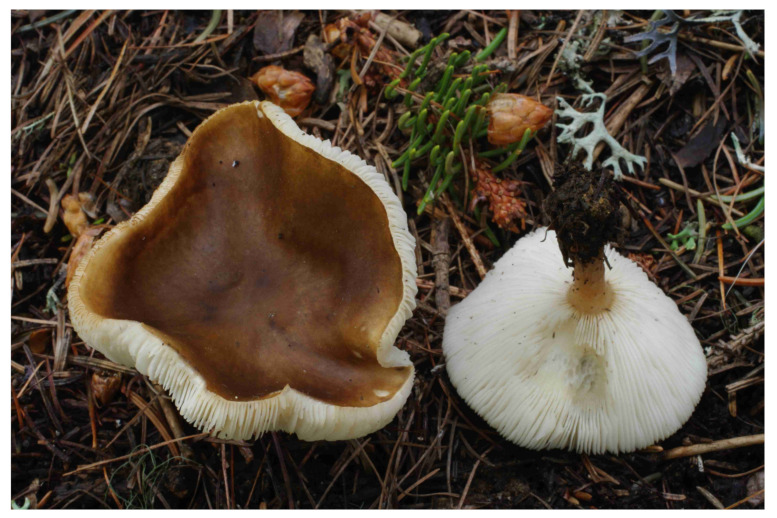
*Melanoleuca acystidiata* (BRNM 772203). Switzerland, Davos, Schatzalp, 25 June 2009, photo J. Borovička.

**Figure 8 jof-07-00191-f008:**
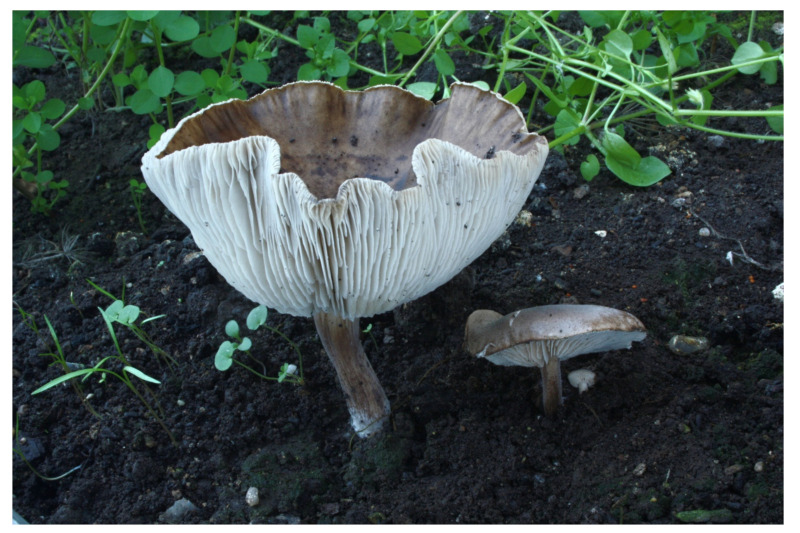
*Melanoleuca castaneofusca* (BRNM 761901). Czech Republic, Hořovice, Osek, Vystrkov, 12 November 2012, photo O. Jindřich.

**Figure 9 jof-07-00191-f009:**
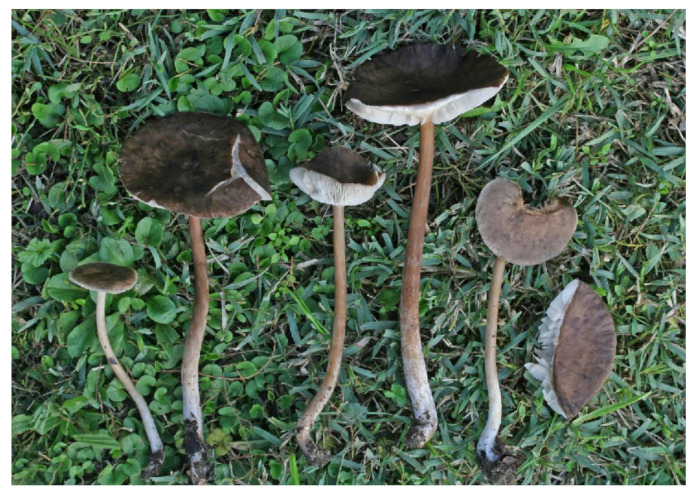
*Melanoleuca luteolosperma* (ANC M0194). Italy, Ferrara, Bosco Spada di Codigoro, 14 November 2008, photo R. Para.

**Figure 10 jof-07-00191-f010:**
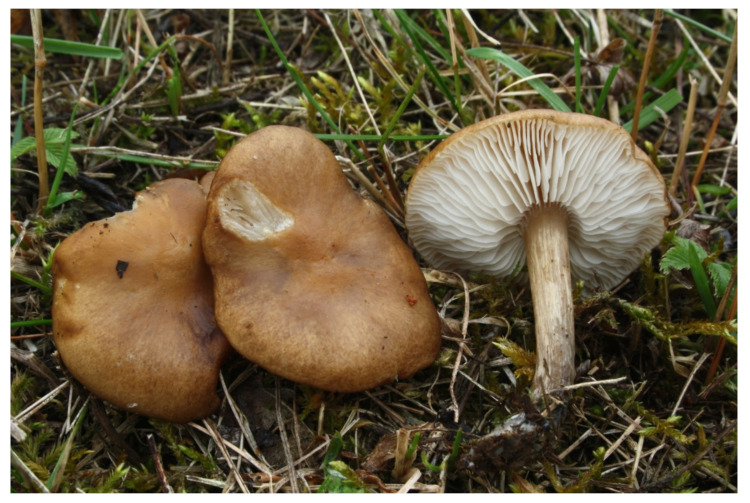
*Melanoleuca luteolosperma* (BRNM 817820). Czech Republic, Koda, 28 October 2009, photo O. Jindřich.

**Figure 11 jof-07-00191-f011:**
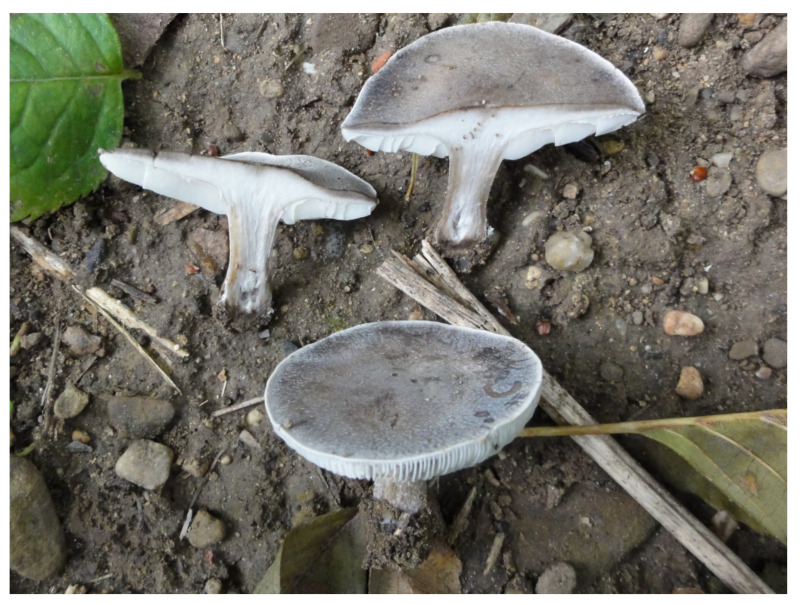
*Melanoleuca luteolosperma* (SLO 1632). Slovakia, Skalka nad Váhom, 10 October 2013, photo O. Ďuriška.

**Figure 12 jof-07-00191-f012:**
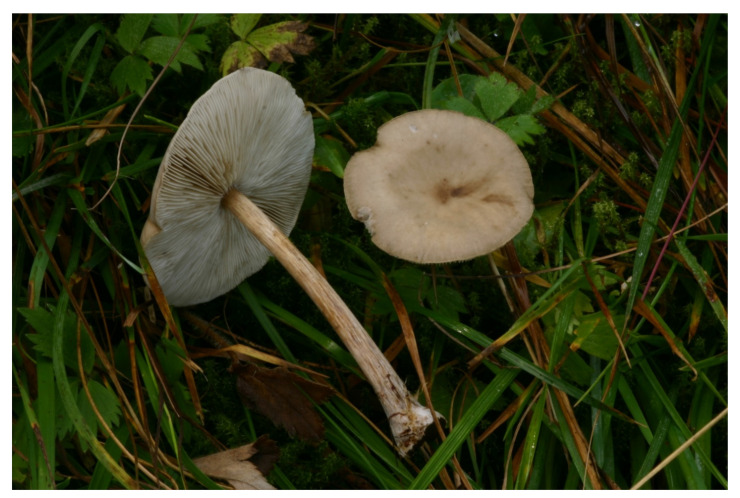
*Melanoleuca microcephala* (BRNM 761890). Slovakia, Ružomberok, Smrekovica, Skalná Alpa, 23 Sept. 2009, photo V. Antonín.

**Figure 13 jof-07-00191-f013:**
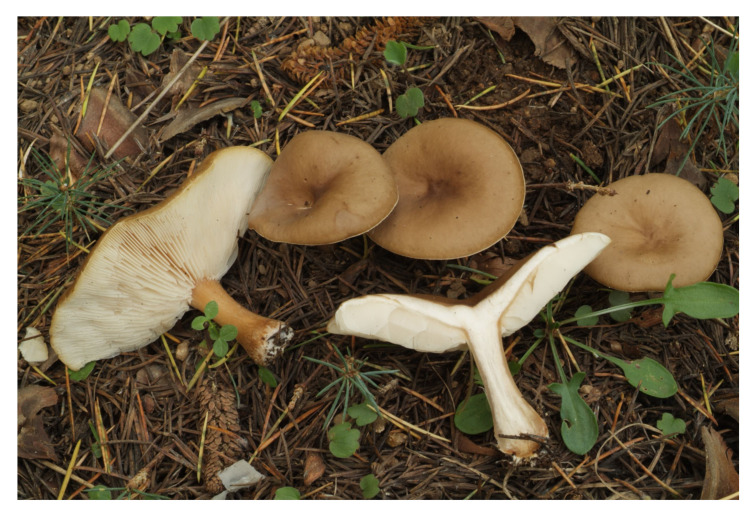
*Melanoleuca paedida* (ANC M0189). Italy, Colamauci di Celico, 16 November 2009, photo C. Lavorato.

**Table 1 jof-07-00191-t001:** The sequenced *Melanoleuca* specimens analyzed in the study. The newly obtained sequences are marked in bold.

Species	Country, Locality	Herbarium Specimen	Genbank acc. no. (ITS)	Genbank acc. no. (tef1)	Genbank acc. no. (rpb2)
*M. acystidiata*, sp. nov.	Italy, South Tyrol, St. Jacob in Val di Vizze	ANC M0205 holotype	JN616462		
*M. acystidiata*, sp. nov.	Switzerland, Davos, Schatzalp	BRNM 772203	**MW491319**	**MW488154**	**MW488169**
*M. angelesiana*	Italy	ANC M0203	JN616420		
*M. angelesiana*	USA, Washington, Olympic Mts., Lake Angels	AFS 11438, syntype	**MW491318**		
*M. castaneofusca*	Italy, Ravenna, Pineta di S. Vitale	BRNM 761900	**MW491323**	**MW488155**	**MW488170**
*M. castaneofusca*	Czech Republic, Hořovice, Osek	BRNM 761901	**MW491320**	**MW488156**	**MW488171**
*M. castaneofusca*	Slovakia, Bratislava	SLO 1639	**MW491324**		
*M. castaneofusca*	Italy, Sardinia, Cagliari	CAG 921103-01	**MW491325**		
*M. castaneofusca*	France, Landrellec	LIP RC86021	**MW491322**		
*M. castaneofusca*	UK, England, Surrey, Kew, Royal Botanic Gardens	K(M)92562	**MW491321**	**MW488157**	**MW488172**
*M. cognata*	Sweden, Västergötland, Trollhättan	GB65454	JX429190		
*M. diverticulata*	Slovakia, Bratislava, Lamač	SLO 1566	LT594155	LT594172	LT594188
*M. exscissa*	Sweden, Västergötland, Trollhättan	GB65455, epitype	JX429192		
*M. exscissa*	Hungary, Bátorliget	BRNM 772198	LT594125	LT594173	LT594189
*M. exscissa*	Czech Republic, Mokrsko	BRNM 781061	LT594122	LT594175	LT594191
*M. exscissa*	Italy, Ravenna, Pineta di S. Vitale	BRNM 781066	LT594123	LT594174	LT594190
*M. fontenlae*, sp. nov.	Italy, Ravenna district, Lido di Dante	BRNM 772194 holotype	**MW491326**	**MW488158**	**MW488173**
*M. fontenlae*, sp. nov.	Slovakia, Cerová vrchovina Mts., Vlčia dolina	SAV F-3823	**MW491327**	**MW488159**	**MW488174**
*M. fontenlae*, sp. nov.	Italy	ANC M0198	JN616460 (as *M. pseudopaedida*)		
*M. friesii*	Slovakia	BRNM 817799	**MT270866**	**MT268581**	**MT268606**
*M. galbuserae*, sp. nov.	Italy, Trentino, Pozza di Fassa	MCVE4505, E. Bizio 1994-08-06	JF908351	**MW488160**	**MW488175**
*M. galbuserae*, sp. nov.	South Tyrol, Solda	BRNM 825710	**MW491333**	**MW488161**	**MW488176**
*M. galbuserae*, sp. nov.	South Tyrol, Solda	BRNM 825709, holotype	**MW491332**		
*M. grammopodia*	Czech Republic, Třemošnice	BRNM 762047	KT279047	KT279048	KT279059
*M. grammopodia*	Slovakia, Veľká Fatra Mts., Vrchlúky	SLO 1463	KP192264	KT279049	KT279058
*M. grammopodia*	Slovakia, Liptovské Revúce	SLO 1468	KP192267	KT279051	KT279061
*M. grammopodia*	Slovakia, Liptovské Revúce	SLO 1466	KP192269	KT279050	KT279060
*M. grammopodia*	Slovenia	ANC M0219	JN616441		
*M. griseobrunnea*	South Korea, Yangpyeong, Pyongchang	BRNM 714980	LT594151		
*M. griseobrunnea*	South Korea, Taean Peninsula, Deoksung	BRNM 781058	LT594152	LT594165	LT594179
*M. humilis*	Czech Republic, Přerov	BRNM 751965	KJ425530	KJ425543	KT279057
*M. humilis*	Czech Republic, Kroměříž	BRNM 710023	KJ425531	KJ425544	KT279055
*M. humilis*	Czech Republic, České Budějovice	CB	KP192290	KT279052	KT279056
*M. juliannae* var. *juliannae*	Hungary, Örkeny	BRNM 751958	KJ425538		
*M. juliannae* var. *juliannae*	Hungary, Budapest, Rákospalota	BP 104371, holotype	KJ425539	KJ425552	LT594182
*M. juliannae* var. *decolorans*	Italy, Altino di Montemonaco	BRNM 751960, holotype	KJ425532	KJ425545	LT594181
*M. leucopoda*	China	HMAS 267626	KF220638		
*M. longisterigma*	Mexico, Veracruz	ENCB, Guzmán 19274	JX429211		
*M. luteolosperma*	Slovakia, Velká Fatra Mts., Lubochňa	BRNM 761907	**MW491328**	**MW488162**	**MW488177**
*M. luteolosperma*	Czech Republic, Srbsko,	BRNM 817820	**MW491329**	**MW488163**	**MW488178**
*M. luteolosperma*	Slovakia, Biele Karpaty, Skalka nad Váhom	SLO 1632	**MW491331**		
*M. luteolosperma*	Germany, Augsburg, Wittelsbacher Park	M 0139486, epitype	**MW491330**		
*M. malenconii*	Czech Republic, Roudnice nad Labem	BRNM 762051	KP192275	KT279053	KT279062
*M. malenconii*	Slovakia, Bratislava	SLO 1455	KP192277	KT279054	KT279063
*M. microcephala*	Slovakia, Velká Fatra Mts., Ružomberok, Skalná Alpa	BRNM 817787	**MW491334**	**MW488164**	**MW488179**
*M. microcephala*	Slovakia, Velká Fatra Mts., Ružomberok, Skalná Alpa	BRNM 817788	**MW491335**	**MW488165**	**MW488180**
*M. microcephala*	Italy, South Tyrol, Franadega di Dobbiaco	ANC M0196	JN616449		
*M. paedida*	Italy, Lombardy, Lago Cancano di Valfurva	R. Para 010907-02	**MW491337**	**MW488166**	**MW488181**
*M. paedida*	Italy, Calabria, Colamauci di Celico	ANC M0189	JN616452		
*M. porphyropoda*	China	HMAS 267624, holotype	KF220640		
*M. pseudoluscina*	Italy	ANC M0191	JN616455		
*M. pseudopaedida*	France, Somme	LIP, Bon 831105, holotype	**MW491336**		
*M. rasilis*	Italy, Monti Sibillini National Park	BRNM 751967	LT594154	LT594171	LT594187
*M. robertiana*	France, Jura, Champagnole	LIP 72092034, holotype	MW491341		
*M. stepposa*	Czech Republic, Ivančice	BRNM 781064	LT594150	LT594162	LT594176
*M. stepposa*	Czech Republic, Brno	BRNM 781099	LT594147	LT594163	LT594177
*M. strictipes*	Czech Republic, Staré Hamry	BRNM 737301	KY417098	MT268561	MT268613
*M. stridula*	Slovakia, Podskalie	BRNM 825716	**MW491340**		
*M. stridula*	Austria, Ehrwald	BRNM 825717	**MW491338**	**MW488167**	**MW488182**
*M. stridula*	Slovakia, Liptovský Hrádok, Hybe	SLO 1543	**MW491339**	**MW488168**	**MW488183**
*M. stridula*	Italy	ANC M0007, neotype	JN616467		
*M. tristis*	Czech Republic, Třeboň	BRNM 772197	LT594137	LT594168	LT594184
*M. tristis*	Italy, Ravenna, Pineta di S. Vitale	BRNM 772192	LT594135	LT594167	LT594183
*M. tristis*	Slovakia, Lakšárska Nová Ves	SLO 1607	LT594139	LT594169	LT594185
*M. tristis*	Slovakia, Šaštín	SLO 1671	LT594140	LT594170	LT594186
*M. verrucipes*		AFTOL-ID 818	DQ490642		
*M. verrucipes*	Switzerland	MCVE 9962	JF908354		
*M. zaaminensis*	Uzbekistan, Pamiro-Altai Mts., Kulsai	TAAM 121360, holotype	LT594141		
*Melanoleuca* sp. Czechia	Czech Republic, Ivančice	BRNM 781065	LT594142	LT594164	LT594178
*Melanoleuca* sp. Korea	South Korea, Mongsanpo	BRNM 781059	LT594153	LT594166	LT594180

## Data Availability

Publicly available datasets were analyzed in this study. This data can be found here: https://www.ncbi.nlm.nih.gov, accessed on 7 January 2021; https://www.mycobank.org, accessed on 7 January 2021.

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
