# Peer review of "Melanoleuca galbuserae, M. fontenlae and M. acystidiata—Three New Species in Subgenus Urticocystis (Pluteaceae, Basidiomycota) with Comments on M. castaneofusca and Related Species"

_jof, 2021, doi:10.3390/jof7030191_

Round 1

Reviewer 1 Report

Here is the review of the paper entitled "Melanoleuca galbuserae, M. fontenlae and M. acystidiata, three new species in subgenus Urticocystis (Pluteaceae, Basidiomycota) with comments to M. castaneofusca and related species" written by Vladimir Antonin & colleagues.

The aim of the paper is to resolve the taxonomic problems in Melanoleuca castaneofusca group of subgen. Urticocystis. Three species (Melanoleuca galbuserae, M. fontenlae, M. acystidiata) are described as new to science based on morphological and molecular characters (multilocus phylogenetic analysis of three marker genes; ITS, rpb2, tef1-alpha). Moreover, the type specimens and recent collections of several related species, namely M. angelesiana, M. castaneofusca, M. luteolosperma, M. pseudopaedida and M. robertiana were analyzed its taxonomy has been clarified. The related species, M. microcephala and M. paedida were included in the analysis.

Generally, the paper analyzed the taxonomy of the pretty difficult group of agarics using the integration of morphological and molecular methods which should be commended. The study is conducted mostly well. The only important drawback of the paper is the lack of a dichotomous key for species identification in this group. The authors should incorporate the key in the revised version to ease the identification of the species within the M. castaneofusca/galbuserae group. 

The English language used should is not of good quality and should be properly revised. Suggested corrections/additions are included in the attached review of the manuscript file.

Best,

Reviewer

Author Response

Dear Editor,

The authors thank to both reviewers for valuable comments and proposals. Both reviews enabled us to improve our manuscript.

All our changes and corrections in the manuscript are made using the "Track Changes" function in Microsoft Word. 

Reviewer 1

The only important drawback of the paper is the lack of a dichotomous key for species identification in this group. The authors should incorporate the key in the revised version to ease the identification of the species within the M. castaneofusca/galbuserae group. 

Answer: There are several taxonomically unclear and insufficiently known taxa among the European species of Melanoleuca, subgenus Urticocystis. These remaining species will be objects of our further studies. Therefore, we do not have data to complete key to identification of all species of the subgenus Urticocystis. The key will be included in our future manuscript, which will close our dealing with this subgenus.

The English language used should is not of good quality and should be properly revised.

Answer: The English was properly revised.

Suggested corrections/additions are included in the attached review of the manuscript file.

Answers (line numbering is according to the original reviewed manuscript):

  1. 29: The global outline by He et al. (2019) was added to the text and references.
  2. 33: The citation [reference number 4] was added at the end of the sentence.
  3. 48: A full stop has been added to the end of this sentence, replacing a comma there.
  4. 146, Table 1: Sequences newly obtained in this manuscript were marked in bold and all specific names are in italics.
  5. 173: The citation “Bon 1991” was replaced by a reference number [10].
  6. 234: The subtitle numbering (3.1.1.) was added.
  7. 249: The coordinates of the type specimen were added.
  8. 347: The term “macrocystidia” was replaced with “cheilocystidia”. This is more proper. There were more cheilocystidia (but only one macrocystidium in caulohymenium) found in the type revision.
  9. 379: The coordinates of the type specimen were added.
  10. 447: This part of the description was changed to “pileus cuticle exceeding the pileus margin“
  11. 484: The citation “Kibby 2016” was replaced by a reference number [35].
  12. 617: The colours “yellow or pale grey brown” belong to the stipe, “pale brown” to the context. It was corrected to be clearer.

All other smaller, mostly language corrections and wording, proposed in the manuscript, were accepted.

Reviewer 2 Report

This paper is a continuation of information on Melanoleuca by authors familiar with the genus. It is valuable contribution to our knowledge of this genus, and describes 3 new species, and analyzes the clade around M. castaneofusca, M. luteolosperma, M. paedida, and M. microcephala molecularly.

Strengths: The Introduction, Materials and Methods, Results, and Taxonomy are basically fine. There were some minor omissions in the taxonomic part that I noted on the manuscript. I tried to smooth out the language in a few places throughout the manuscript as well.

Weaknesses:  I did not think that the Abstract reflected the research and appeared rather disjointed and scattered, and should be rewritten to reflect what was done. Start with mentioning the subgenus Urticocystis as an umbrella. Then use the nice, relevant sentence at the very end of the intro. Then give the results.

Some details:  Sometimes the trama was described as inamyloid and sometimes non-dextrinoid, which was confusing so be consistent on this.  Which is best?  Some of the Diagnoses were not consistent in form, see my notes on the manuscript.  I do think caulohymenium should be defined somewhere and there is a perfect opening for this in the discussion. I am not sure about the journal’s requirement for species authorities, but many were missing from the text. Alternatively, they could all be placed in the table, but they need to be somewhere in a taxonomic paper. The descriptions in the diagnosis overlapped strongly with the first part of each Remark section, so if the paper needs to be pared down, this should be considered. Did the authors want to be more consistent in saying long versus short stipe (shorter than pileus width) or is that not possible? There were 3 photos of M. luteolosperma and they all looked like very different taxa? This should be mentioned and accounted for. Check to see if Fig 11 is labeled correctly. 

It was disappointing that there were no microdrawings of spores and cystidia, but given that these are not macrocystidia, I guess it is ok. There was only one microphotograph of spores which does not appear to be that useful, yet 3 new species were described. I guess that is ok, given the subgenus. A KEY(!) for the M. castaneofusca group would really improve the paper!  Then the verbal comparisons would make more sense, or would be unnecessary.  It would be a great contribution to this part of the genus.

Again, I made a lot of small edits on the manuscript itself which I hope are helpful.  It is a worthwhile paper on a difficult genus and should be published.   

Author Response

Dear Editor,

The authors thank to both reviewers for valuable comments and proposals. Both reviews enabled us to improve our manuscript.

All our changes and corrections in the manuscript are made using the "Track Changes" function in Microsoft Word. 

Reviewer 2

I did not think that the Abstract reflected the research and appeared rather disjointed and scattered, and should be rewritten to reflect what was done. Start with mentioning the subgenus Urticocystis as an umbrella. Then use the nice, relevant sentence at the very end of the intro. Then give the results.

Answer: The Abstract was corrected starting with more general sentences as proposed.

Sometimes the trama was described as inamyloid and sometimes non-dextrinoid, which was confusing so be consistent on this.  Which is best? 

Answer: The terms “non-dextrinoid” were replaced with “inamyloid” throughout the manuscript.

Some of the Diagnoses were not consistent in form, see my notes on the manuscript. 

Answer: They were corrected. All diagnoses are written in the same form now.

I do think caulohymenium should be defined somewhere and there is a perfect opening for this in the discussion.

Answer: The definition of the caulohymenium was added to Materials and Methods, Morphological dataset (line 65 in the corrected manuscript) and the citation to References.

I am not sure about the journal’s requirement for species authorities, but many were missing from the text. Alternatively, they could all be placed in the table, but they need to be somewhere in a taxonomic paper.

Answer: Species authorities are included in the first appearing of the taxon. The omissions were corrected.

The descriptions in the diagnosis overlapped strongly with the first part of each Remark section, so if the paper needs to be pared down, this should be considered.

Answer: Diagnoses were corrected and differ from the Remark section now.

Did the authors want to be more consistent in saying long versus short stipe (shorter than pileus width) or is that not possible?

Answer: The parameters of the stipe length versus the pileus diameter were added to all descriptions.

There were 3 photos of M. luteolosperma and they all looked like very different taxa? This should be mentioned and accounted for. Check to see if Fig 11 is labelled correctly.

Answer: Melanoleuca luteolosperma is extremely variable in colour, macroscopically sometimes reminding of different species. Figure 11 is labelled correctly. The identity of this collection is also confirmed phylogenetically (see Table 1).

It was disappointing that there were no microdrawings of spores and cystidia, but given that these are not macrocystidia, I guess it is ok.

Answer: Basidiospores and cystidia of studied species are very similar. All drawing would be almost identical and non-bringing closer view to microscopic characters. Characteristic shapes of cystidia (exscissa- or brevipes-type) are defined by Vizzini et al. 2011 (reference number [4] in the text), where the types of cystidia are drawn and clearly described.

There was only one microphotograph of spores which does not appear to be that useful, yet 3 new species were described. I guess that is ok, given the subgenus.

Answer: We propose to keep this photo in the manuscript even if it is the only one. Melanoleuca galbuseare belong to different species group and, by our opinion, it has sense to publish it here for comparison to closely related M. stepposa, M. tristis and M. zaaminensis.

A KEY(!) for the M. castaneofusca group would really improve the paper!  Then the verbal comparisons would make more sense, or would be unnecessary.  It would be a great contribution to this part of the genus.

Answer: Please, see the first answer to the Reviewer 1.

I made a lot of small edits on the manuscript itself which I hope are helpful.  It is a worthwhile paper on a difficult genus and should be published.  

Answers (line numbering is according to the original reviewed manuscript):

  1. 4: The language correction was made.
  2. 173: The meaning of the finger text was briefly explained in the Materials and Methods, Morphological dataset chapter, and the citation is added to References.
  3. 176: The number of lamellae and lamellulae was added.
  4. 304: The identity of Melanoleuca fontenlae with M. pseudopaedida sensu Vizzini et al. was confirmed phylogenetically.
  5. 406: The using of “alpine” was replaced with “montane” here.
  6. 415: Corrected to “shorter or longer than the pileus width“.
  7. 618: The size of basidiospores was added.
  8. 629: A short conclusion was added at the end of this paragraph.
  9. 639: The end of this sentence was re-written.
  10. 690: This species was collected in the montane and alpine altitudinal belt in various habitats, e.g. in an alpine meadow, in spruce forest etc.
  11. 758: The information about the lamellar trama was added.

All other smaller, mostly language corrections and wording, proposed in the manuscript, were accepted.

Round 2

Reviewer 1 Report

Dear authors and the editor,

I find the revised version of the paper suitable for publication in Journal of Fungi. No further corrections are needed.

Best, Reviewer

Author Response

Dear Reviewer,

many thanks for your very valuable review in the first round once more. It helped us to improve our manuscript very much.

Best regards,

Vladimir Antonin

coresponding author